

# Uncertainty of simulated brightness temperature due to sensitivity to atmospheric gas spectroscopic parameters

Donatello Gallucci[1], Domenico Cimini[1,2], Emma Turner[3], Stuart Fox[3], Philip W. Rosenkranz[4], Mikhail Y. Tretyakov[5], Vinia Mattioli[6], Salvatore Larosa[1], and Filomena Romano[1]

[1]National Research Council of Italy, Institute of Methodologies for Environmental Analysis, Potenza, 85050, Italy
[2]Center of Excellence CETEMPS, University of L'Aquila, L'Aquila, 67100, Italy
[3]Met Office, FitzRoy Road, Exeter, EX1 3PB, UK
[4]Massachusetts Institute of Technology, Cambridge, MA 02139, USA
[5]Russian Academy of Sciences, Institute of Applied Physics, Nizhny Novgorod, 603950, Russia
[6]European Organisation for the Exploitation of Meteorological Satellites, Darmstadt, Germany

**Correspondence:** Donatello Gallucci (donatello.gallucci@cnr.it); Domenico Cimini (domenico.cimini@cnr.it)

**Abstract.** Atmospheric radiative transfer models are extensively used in Earth observation to simulate radiative processes occurring in the atmosphere and to provide both upwelling and downwelling synthetic brightness temperatures for ground-based, airborne, and satellite radiometric sensors. For a meaningful comparison between simulated and observed radiances, it is crucial to characterise the uncertainty of such models. The purpose of this work is to quantify the uncertainty in radiative transfer models due to uncertainty in the associated spectroscopic parameters, and to compute simulated brightness temperature uncertainties for millimeter- and submillimeter-wave channels of downward-looking satellite radiometric sensors (MWI, ICI, MWS and ATMS) as well as upward looking airborne radiometers (ISMAR and MARSS). The approach adopted here is firstly to study the sensitivity of brightness temperature calculations to each spectroscopic parameter separately, then to identify the dominant parameters and investigate their uncertainty covariance, and finally to compute the total brightness temperature uncertainty due to the full uncertainty covariance matrix for the identified set of relevant spectroscopic parameters. The approach is applied to a recent version of the Millimiter-Wave propagation model, taking into account water vapor, oxygen, and ozone spectroscopic parameters, though it is general and can be applied to any radiative transfer code. A set of 135 spectroscopic parameters were identified as dominant for the uncertainty of simulated brightness temperatures (26 for water vapor, 109 for oxygen, none for ozone). The uncertainty of simulated brightness temperatures is computed for six climatology conditions (ranging from sub-Arctic winter to Tropical) and all instrument channels. Uncertainty is found to be up to few kelvin [K] in the millimeter-wave range, whereas it is considerably lower in the submillimeter-wave range (less than $1\,\mathrm{K}$).

## 1 Introduction

Radiative transfer models (RTM) are widely used to compute the propagation of electromagnetic radiation through the Earth's atmosphere and to simulate radiometric observations of natural radiation (Rosenkranz ((1993))). At the core of RTM are atmospheric absorption models, which simulate the absorption/emission of electromagnetic radiation by atmospheric constituents. RTM represents the forward operator for atmospheric radiometric applications. Thus, RTM are widely exploited for the solu-





tion of the inverse problem, i.e. the retrieval of atmospheric parameters from radiometric observations (Rodgers (2000)), and for data assimilation of radiometric observations in numerical weather prediction (NWP) models (Saunders et al. (2018)). In addition, as part of their quality control, radiometric observations from satellites are often validated against simulated radiances

obtained by processing thermodynamic profiles from radiosondes or NWP models with RTM (Clain et al. (2015), Kobayashi et al. (2017)). Therefore, RTM and absorption models have general application for atmospheric sciences, including meteorology and climate studies. All these applications would benefit from a careful characterization of RTM uncertainty. For example, instrument validation through comparison of observations and simulations should consider the uncertainty of both to be metrologically meaningful (Bodeker et al. (2016), Yang et al. (2023)). However, the characterisation of uncertainty associated with

simulated brightness temperatures is generally lacking within the scientific literature. This work aims to fill this gap, providing a thorough analysis of the uncertainty of simulated brightness temperatures due to assumptions in the atmospheric absorption model.

Synthetic brightness temperatures ($T_\mathrm{B}$) simulated with atmospheric radiative transfer and absorption models are inherently affected by uncertainty, due to the assumed values for the intrinsic spectroscopic parameters. These values are in fact deter-

mined either from theoretical calculations, lab experiments or field measurements, and are thus affected by either computational or experimental uncertainty. This uncertainty then propagates from the spectroscopic parameters through the absorption model and RTM calculations, and finally to simulated $T_\mathrm{B}$ and atmospheric retrievals. It is therefore crucial to provide an estimate of the $T_\mathrm{B}$ uncertainty value, in order to have an adequate interpretation of the observation-simulation statistics, and to fulfil international standards requirements.

Therefore, the rationale for this work is to fully characterize the synthetic $T_\mathrm{B}$ uncertainty due to the uncertainty of atmospheric gas spectroscopic parameters, following the approach proposed by Cimini et al. (2018). In particular, the scope is to assess the uncertainty of the synthetic brightness temperatures obtained via the Millimiter-wave Propagation model based on the spectroscopy from Rosenkranz et al. (2018). The approach consists of mapping the uncertainty of the $T_\mathrm{B}$ to each single spectroscopic parameter. The analysis is performed in four steps: i) review open literature concerning spectroscopic parame-

ters relevant for the frequency range of interest (16-700 GHz) for assessing the associated uncertainties (this can be found in Turner et al. (2022)); ii) perform a sensitivity study to investigate the dominant uncertainty contribution to radiative transfer calculations; iii) estimate the full uncertainty covariance matrix for the reduced set of dominant parameters; iv) propagate the uncertainty covariance matrix to estimate the impact on simulated brightness temperatures. We perform the above analysis for the estimation of the uncertainty on simulated brightness temperature in the frequency range $16 - 700$ GHz, both for the

downward-looking view at $53°$ from nadir at the top-of-atmosphere (TOA) - i.e., the observation geometry of the EUMETSAT Polar System Second Generation (EPS-SG) MicroWave (MWI) and Ice Cloud (ICI) Imagers - and for the zenith upward looking view from different heights, as feasible for airborne sensors. The estimated uncertainty spectra are also convolved on the finite channel bandwidths of the relevant satellite and airborne instruments. For the downward looking geometry we consider MWI and ICI, as well as the EPS-SG MicroWave Sounder (MWS) and the Advanced Technology Microwave Sounder

(ATMS) aboard NOAA satellites (Suomi-NPP, NOAA-20, NOAA-21). For the upward looking geometry, we consider selected





channels from the International Submillimetre Airborne Radiometer (ISMAR) (Fox et al., 2017) and the Microwave Airborne Radiometer Scanning System (MARSS) (McGrath and Hewison, 2001).

The motivation for selecting the above frequency range and instruments is explained below. The EPS-SG will contribute with a new generation of polar-orbiting satellites in the timeframe from 2025 onward (Accadia et al., 2020; Mattioli et al., 2019), providing continuity to the current EUMETSAT EPS programme. For the EPS-SG a number of missions have been identified, which include the aforementioned MWI, ICI and MWS missions. This study is indeed in preparation and support of the Cal/Val activities and exploitation of these missions, and focuses on the quantitative assessment of atmospheric absorption model uncertainty in the frequency range encompassing the instruments channels of interest (i.e. $16-700\,\text{GHz}$ ). The outcome of this study will also be applicable to the ATMS mission, as a representative from the MW/heavily used instrument in current operation. The airborne instruments are used as demonstrators for EPS-SG.

MWI and ICI are two conically scanning microwave radiometers. MWI will have 18 channels ranging from 18 to 183 GHz, providing continuity of key microwave imager missions. Four channels at 18.7, 23.8, 31.4 and 89 GHz provide key information on weather forecasting, as well as precipitation, total column water vapour and cloud liquid water. MWI also includes a new set of channels near 50–60 GHz and at 118 GHz, allowing retrieval of weak precipitation and snowfall. ICI is instead specifically designed to support remote sensing of cloud ice, and constitutes a novelty of this kind. ICI frequencies will cover the mm/sub-mm range spectrum from 183 GHz and 664 GHz: eleven channels in the water vapour absorption lines (i.e., 183, 325 and 448 GHz) whereas 243 and 664 GHz in atmospheric windows. ICI information on humidity and ice hydrometeors will be crucial to characterise clouds properties. The rotation of the slanted antennas allows conical scans with constant incidence angles of about 53°, depending on the channel frequency. MWS is a cross-track scanning radiometer. MWS will comprise 24 channels from 23.8 to 229 GHz. The 14 oxygen-band channels near 50-60 GHz provide microwave temperature sounding, while the water vapour channel at 23.8 GHz and the five channels at 183.31 GHz are used for humidity retrievals. The instrument also carries a new channel at 229 GHz. Both the microwave sounders MWS and ATMS provide information about thermodynamics of the atmosphere, such as temperature and moisture profiles. The microwave sounders MWS and ATMS are both based on a cross-track sensing mechanism, so that the Earth is observed at different scanning angles, symmetric around the nadir direction, with an angular sampling spaced by 1.05° and a maximum scanning angle of 49.31°.

To the best of our knowledge, this is the first study investigating the characterisation of synthetic upwelling $T_B$ uncertainty due to the sensitivity of gas spectroscopic parameters. Moreover, it extends the work of (Cimini et al., 2018, 2019) providing downwelling $T_B$ uncertainty at different heights and to a wide range of frequencies covering from microwave to millim.-wave $16-700\,\text{GHz}$. Although this study adopts the same underlying approach as in (Cimini et al., 2018, 2019), it differs in the i) viewing geometry (satellite/airborne vs. ground-based), ii) absorption model (featuring new spectroscopy, with additional parameters being investigated), and iii) frequency range, extended by one order of magnitude. Note that a thorough characterization of the uncertainty affecting the simulated brightness temperatures implies better understanding of their limitations when used for the training of inverse algorithms, the monitoring of sensor calibration, and the data assimilation of real observations into NWP models.





The paper is organized as follows: Section 2 introduces the theoretical basis and reports on the absorption model sensitivity analysis to spectroscopic parameters; Section 3 discusses the implications of spectral channel convolution; Section 4 reports on the estimation of the full uncertainty covariance matrix for the spectroscopic parameters; Section 5 presents the results of the uncertainty propagation from spectroscopic parameters to simulated $T_{\mathrm{B}}$; finally, Section 6 presents a summary and draws final conclusions.

## 2   Sensitivity Analysis

In this section we briefly introduce the theoretical basis underlying the calculation of the modelled brightness temperature uncertainty, propagating the spectroscopic parameters' uncertainty into the simulated brightness temperature following the method outlined in Cimini et al. (2018). The method first performs a review of the spectroscopic parameters and their uncertainty and then a sensitivity analysis to identify the dominant contributions.

This study exploits a state-of-the-art microwave radiative transfer model, applicable to airborne as well as ground-based and satellite observation geometries. We will adopt the Millimeter-wave Propagation Model using the atmospheric absorption equations by (Rosenkranz ((1993)), with updated spectroscopic parameters, which will be referred to as PWR19 (see also Larosa et al. (2023) for code implementation). The brightness temperature simulated with this model is generally a function of the spectroscopic parameters considered within the model. Under the assumption of small perturbations, non-linear dependence

can be reasonably linearized as:

$$T_{\mathrm{B}} = K_p \cdot (p - p_0) + T_{\mathrm{B},0} \tag{1}$$

where $p$ is a vector whose elements are the parameters in the model, with nominal value $p_0$, $T_{\mathrm{B}}$ is a vector of calculated brightness temperatures at various frequencies using parameter values $p$, while $T_{\mathrm{B},0}$ is calculated for parameter values $p_0$, and $K_p$ represents the model parameter Jacobian, i.e. the matrix of partial derivatives of model output with respect to model

parameters $p$.

The approach adopted here to compute the $T_{\mathrm{B}}$ uncertainty due to the uncertainties of all gas spectroscopic parameters within the model consists firstly in identifying the dominant parameters causing the uncertainty, so to reduce the dimensionality of the problem. Hence, we investigate the sensitivity of the model to each spectroscopic parameter, separately, by perturbing the value of that parameter by its estimated uncertainty: if the sensitivity is above a given threshold, the parameter is deemed relevant

and considered for further analysis, otherwise it is discarded. We choose to set the threshold equal to $0.1\,\mathrm{K}$, typically below the uncertainty for radiometric observations.

Once we have singled out the reduced set of relevant parameters, the full uncertainty covariance matrix $(\mathrm{Cov}(p))$ is estimated by considering the possible correlations between the spectroscopic parameters. Then, the Jacobian of the radiative transfer model with respect to dominant spectroscopic parameters $(K_p)$ is computed by small perturbation analysis. Finally, indicating

with $^\mathsf{T}$ the matrix transpose, the full uncertainty covariance matrix for the computed brightness temperature is derived from



$\mathrm{Cov}(\boldsymbol{p})$ and $\boldsymbol{K_p}$ as (BIPM et al.):

$$\mathrm{Cov}(\boldsymbol{T_{\mathrm{B}}}) = \boldsymbol{K_p} \cdot \mathrm{Cov}(\boldsymbol{p}) \cdot \boldsymbol{K_p}^{\mathsf{T}} \tag{2}$$

In this work we only consider spectroscopic parameters exploited in PWR19, unless otherwise specified. As anticipated before, the model sensitivity to a given parameter is computed by perturbing the value of that parameter by the estimated uncertainty (at 1-sigma level). Each parameter has been investigated individually by perturbing its value by $\pm\sigma$ (1-sigma) uncertainty and computing the impact on the modelled $T_{\mathrm{B}}$ as the difference between $T_{\mathrm{B}}$ computed with the nominal value of the parameter, and $T_{\mathrm{B}}$ computed with the perturbed value, i.e.:

$$\Delta \boldsymbol{T}_{\mathrm{B}\,\mathrm{i},+/-} = \boldsymbol{T}_{\mathrm{B}}(p_{\mathrm{i}}) - \boldsymbol{T}_{\mathrm{B}}(p_{\mathrm{i}} \pm \sigma_{p_{\mathrm{i}}}) \tag{3}$$

Monochromatic radiative transfer calculations are performed in the 16-700 GHz range at 50 MHz resolution, with the addition of selected frequencies corresponding to central frequency of MWI and ICI (Table 4), MWS and ATMS channels (Table 5). Six different climatology conditions are considered to account for temperature, pressure, and humidity dependence: Tropical, Mid-Latitude Summer, Mid-Latitude Winter, Sub-Arctic Summer, Sub- Arctic Winter and U.S Standard profiles (Anderson et al. (1986)). Thus, for each parameter, both $\boldsymbol{T}_{\mathrm{B}\,\mathrm{i},+}$ and $\boldsymbol{T}_{\mathrm{B}\,\mathrm{i},-}$ are computed for each of the six typical climatology conditions. For downward-looking geometry, the surface emissivity must be modelled to compute $\boldsymbol{T}_{\mathrm{B}\,\mathrm{i},+/-}$. In general we expect the higher the emissivity, the lower the sensitivity to spectroscopic parameters. In fact, a higher emissivity leads to lower contribution of downwelling radiation to the radiation reaching a satellite down-looking instrument, thus the sensitivity to spectroscopic parameters is reduced to the upwelling path only. Since oceans cover about $70\,\%$ of the Earth, the surface emissivity is modelled over water, using the Tool to Estimate Sea- Surface Emissivity from Microwaves to sub-Millimeter waves (TESSEM2,Prigent et al. (2017)). The emissivity is computed at $53°$ from nadir, corresponding to the ICI and MWI observing angle, assuming typical ocean conditions ($8\,\mathrm{m/s}$ wind speed; $290\,\mathrm{K}$ sea surface temperature; $35\,\mathrm{PSU}$ salinity). TESSEM2 provides emissivity at both H and V polarizations, with the emissivity at H-pol lower than at V-pol in the frequency range of interest. Most of MWI and ICI channels are V-pol (except for window channels featuring both H and V); however, for figure clarity, we consider only the most conservative case, i.e., H-pol emissivity (as previously stated, lower emissivity leads to higher sensitivity). So, hereafter figures show simulations obtained with H-pol emissivity, whereas the reader is referred to the tables in Appendix A for comparison between the two polarizations.

The following sections introduce the spectroscopic parameters of the relevant gases in the considered frequency range, i.e. $H_2O$, $O_2$ and $O_3$, with selected examples of the corresponding $\boldsymbol{T}_{\mathrm{B}\,\mathrm{i},+}$ and $\boldsymbol{T}_{\mathrm{B}\,\mathrm{i},-}$ spectra. Other uncertainty sources, such as uncertainties of uncertainties or the uncertainties from minor absorbers/lines are considered as second order contributions; since uncertainty adds up in quadrature, second order contributions add relatively little with respect to first order contributions.

## 2.1 Sensitivity to $H_2O$ parameters

This section investigates the RTM sensitivity to water vapor spectroscopic parameters. In the frequency range under consideration, several resonant lines (from 22 to 916 GHz) contribute non-negligibly to the water vapor absorption. In addition, a





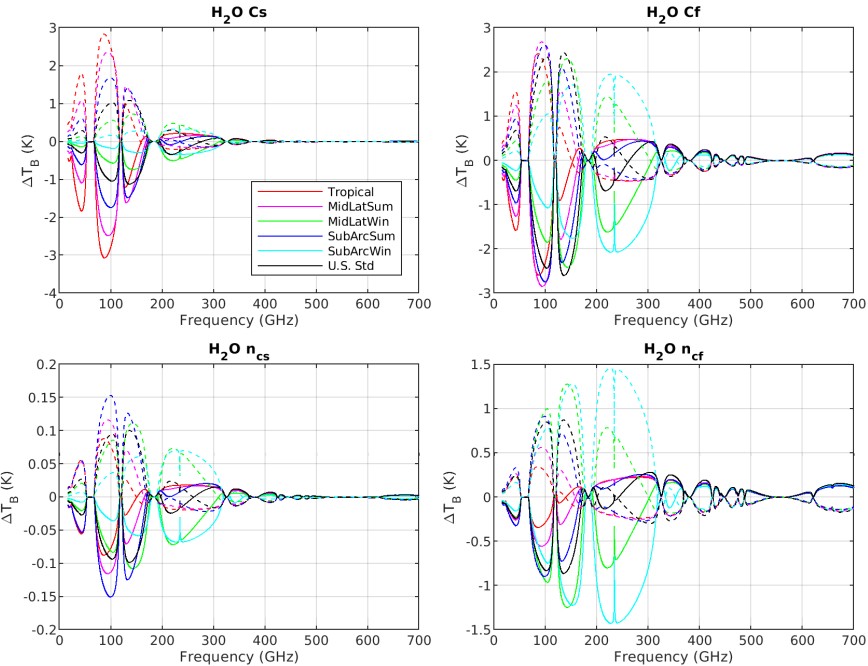

**Figure 1.** Sensitivity of modelled $T_B$ to water vapor continuum absorption parameters, for downward looking geometry at $53°$ from nadir, with H-pol sea surface emissivity. Solid lines correspond to negative perturbation ($\Delta T_{Bi,-}$), while dashed lines to positive perturbation ($\Delta T_{Bi,+}$)). Top: Self- ($C_s$) and foreign- ($C_f$) induced broadening coefficients. Bottom: Self- and foreign-broadening temperature-dependence exponents (respectively $n_{c_s}$ and $n_{c_f}$). Different colors indicate six different climatology conditions.

non-resonant contribution is given by the so-called water vapor continuum absorption. For the resonant absorption, the following parameters are relevant: line frequency ($\nu$), intensity ($S$), the temperature dependence of the partition sum ($n_S$) (i.e., the total number of populated molecular states), the lower-state energy ($E_{low}$), air- and water-broadening ($\gamma_a$ and $\gamma_w$) and their temperature-dependence exponents ($n_a$ and $n_w$), and air- and water shifting ($\delta_a$ and $\delta_w$). For the continuum absorption, four parameters are relevant, namely the self- and foreign-induced intensity coefficients and their respective temperature-dependence exponents ($C_s, C_f, n_{cs}, n_{cf}$). Note that this model for the wv continuum absorption was specifically developed to address the MW range and later extended to higher frequencies. However, more recently new models have been proposed based on measurements and ab-initio calculations to improve the fits in the mm-wave range (Odintsova et al., 2017; Koroleva et al., 2021) However, the dominant foreign continuum fits well the $f^2$ dependence up to 1 THz, as reviewed recently (Koroleva et al., 2021). The water vapor parameters perturbed in the sensitivity analysis are listed in Table 1, together with the references from which their values and estimated uncertainty are derived (i.e., Cimini et al. (2018), Turner et al. (2022) and references therein).

   Figure 1 shows the $\Delta T_{Bi,+/-}$ spectra corresponding to the perturbation of the four parameters used to model the water vapor continuum absorption ($C_s, C_f, n_{cs}, n_{cf}$). Each panel shows the sensitivity of modeled 16- 700 GHz $T_B$ to one parameter



**Table 1.** List of water-vapor parameters perturbed in the sensitivity analysis.

| Symbol (Units) | Parameter | Uncertainty [%] | Reference |
|---|---|---|---|
| $\nu$ (kHz) | Resonant line frequency | $2 \times 10^{-7} - 5 \times 10^{-4}$ | 22 − 183 GHz: Cimini et al. (2018) Other lines: HITRAN database |
| $S$ (Hz cm$^2$) | Resonant line intensity | $1 - 2$ | Turner et al. (2022) (Table 10) |
| $n_s$ (unitless) | Resonant line intensity temperature dependence exponent | $0.5$ | Cimini et al. (2018) and references therein |
| $E_{\text{low}}$ (cm$^{-1}$) | Resonant line lower-state energy | $\sim 10^{-7}$ | Cimini et al. (2018) and references therein |
| $\gamma_{\text{a}}$ (GHz bar$^{-1}$) | Resonant line air-broadening | $0.43 - 5$ | Turner et al. (2022) (Table 10) |
| $\gamma_{\text{w}}$ (GHz bar$^{-1}$) | Resonant line water-broadening | $0.15 - 2.54$ | Turner et al. (2022) (Table 10) |
| $n_{\text{a}}$ (unitless) | Resonant line air-broadening temperature dependence exponent | $0.93 - 14.06$ | Cimini et al. (2018) and references therein Turner et al. (2022) (Table 10) |
| $n_{\text{w}}$ (unitless) | Resonant line water-broadening temperature dependence exponent | $9.46 - 41.67$ | Cimini et al. (2018) and references therein Turner et al. (2022) (Table 10) |
| $\delta_{\text{a}}$ (GHz bar$^{-1}$) | Resonant line air-shifting | $7.12 - 38.01$ | Turner et al. (2022) (Table 10) |
| $\delta_{\text{w}}$ (GHz bar$^{-1}$) | Resonant line water-shifting | $0.04 - 13.02$ | Turner et al. (2022) (Table 10) |
| $C_{\text{f}}$ (km$^{-1}$mb$^{-2}$GHz$^{-2}$) | Foreign-broadened continuum | $9.01\%$ | Turner et al. (2022) (Table 11) |
| $C_{\text{s}}$ (km$^{-1}$mb$^{-2}$GHz$^{-2}$) | Self-broadened continuum | $22.78\%$ | Turner et al. (2022) (Table 11) |
| $n_{c_{\text{f}}}$ (unitless) | Foreign-broadened continuum temperature dependence exponent | $13.33\%$ | Turner et al. (2022) (Table 11) |
| $n_{c_{\text{s}}}$ (unitless) | Self-broadened continuum temperature dependence exponent | $4\%$ | Turner et al. (2022) (Table 11) |

only, as computed for the six climatology conditions. The symmetry of $\Delta T_{\text{B}i,+}$ and $\Delta T_{\text{B}i,-}$ with respect to the zero line suggests that estimated uncertainties represent small perturbations satisfying the linearization assumed above.

Similarly, Figure 2 shows the $\Delta T_{\text{B}i,+}$ and $\Delta T_{\text{B}i,-}$ spectra corresponding to the perturbation of the broadening and shifting parameters used to model the water vapor line absorption ($\gamma_{\text{a}}, \gamma_{\text{w}}, n_{\text{a}}, n_{\text{w}}, \delta_{\text{a}}, \delta_{\text{w}}$), while Figure shows 3 the perturbation of the line intensity ($S$), its temperature dependence ($n_{\text{S}}$), the central frequency ($\nu$), and the lower-state energy ($E_{\text{low}}$). We perturbed




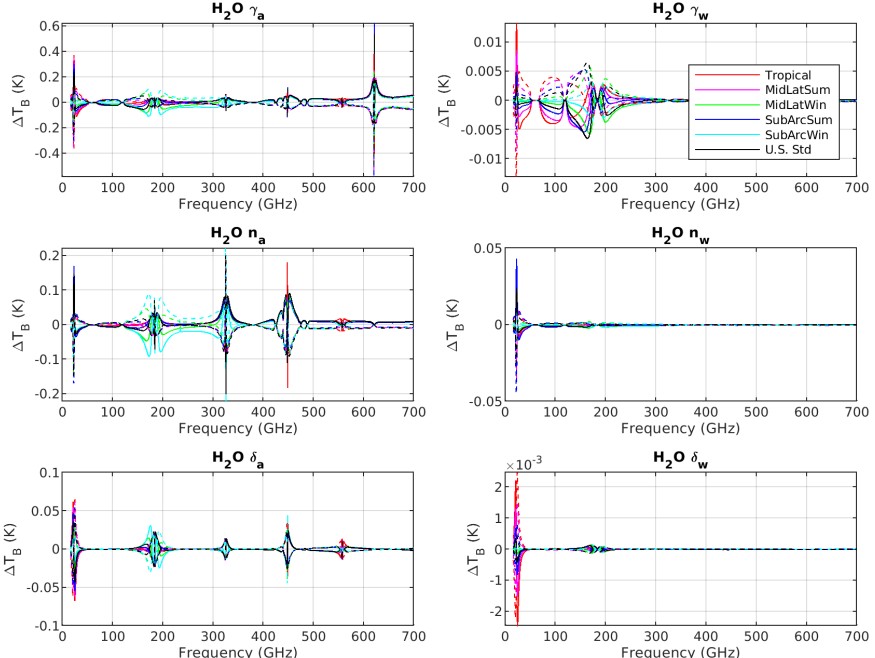

**Figure 2.** As in Fig. 1, but for water vapor line absorption, broadening and shifting parameters. Solid lines correspond to negative perturbation ($\Delta T_{\mathrm{Bi},-}$), while dashed lines to positive perturbation ($\Delta T_{\mathrm{Bi},+}$)). Top: Air- ($\gamma_{\mathrm{a}}$) and water- ($\gamma_{\mathrm{w}}$) induced broadening coefficients. Middle: Temperature-dependence exponents of air- ($n_{\mathrm{a}}$) and water- ($n_{\mathrm{w}}$) induced broadening. Bottom: Air- ($\delta_{\mathrm{a}}$) and water- ($\delta_{\mathrm{w}}$) induced shifting coefficients.

the parameters of the six water-vapor key stronger lines together. If the impact is less than 0.1 for all, then the parameter is discarded. If the impact is higher than 0.1 for any of them, then the parameter is evaluated for each line and only those with impact higher than $0.1\,K$ are retained for further analysis. The sensitivity analysis shows that among the model parameters that were perturbed by the estimated uncertainty (Table 1), only seven types impact the modelled upwelling 16-700 GHz $T_{\mathbf{B}}$

by more than 0.1 K: the four continuum parameters ($C_{\mathrm{s}}, C_{\mathrm{f}}, n_{\mathrm{c_s}}, n_{\mathrm{c_f}}$), and four line parameters ($\gamma_{\mathrm{a}}, n_{\mathrm{a}}, S, \nu$). Among the latter, the central frequency $\nu$ will not be considered for the reasons explained in Section 3. The other three parameters have been considered for six key water vapor lines (i.e.: 22, 183, 325, 448, 556, 752 GHz). In addition, the following line parameters were found relevant ($\Delta T_{\mathbf{B}} > 0.1\,K$):

- $S$ for 380, 474, and 620 GHz;

– $\gamma_{\mathrm{a}}$ for 620 GHz

Therefore, 26 parameters were identified as dominant for $H_2O$ absorption uncertainty, and are further considered for evaluation of their covariance in Section 4.



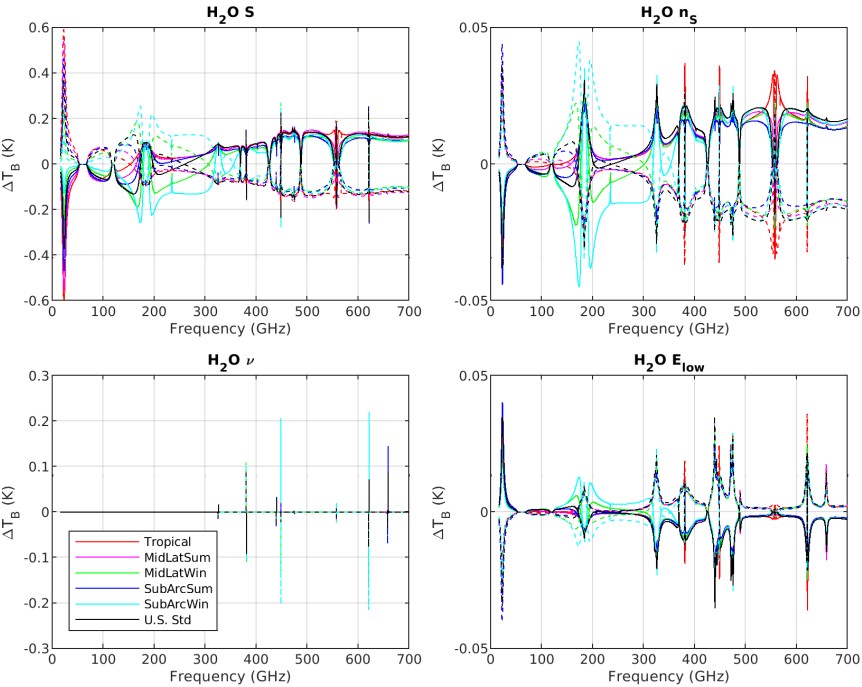

**Figure 3.** As in Fig. 2 but for line intensity ($S$), its temperature dependence ($n_S$), the central frequency ($\nu$), and the lower-state energy ($E_{low}$).

## 2.2 Sensitivity to $O_2$ parameters

Oxygen contributes to the absorption with several resonant lines in the frequency range under consideration. The PWR19 model
includes 49 oxygen absorption lines, of which 37 are within the 60 GHz band, one lies at 118 GHz and the remaining 11 are in
the mm/sub-mm range (200-900 GHz). In addition, the non-resonant contribution is given by a zero-frequency transition, i.e.
the $O_2$ non-resonant contribution is modelled as a pseudo-line at zero-frequency (van Vleck, 1947), as discussed in Cimini et al.
(2018). For the resonant absorption, the following parameters are relevant: line frequency ($\nu$), intensity ($S$) and its temperature
coefficient ($n_S$), the lower-state energy ($E_{low}$), air-broadening ($\gamma_a$) and its temperature-dependence exponent ($n_a$), normalized
mixing coefficient ($Y$) and its temperature-dependence coefficient ($V$), and water-to-air broadening ratio ($r_{\omega 2a}$). For the zero-
frequency absorption, two parameters are relevant, the intensity ($S_0'$) and broadening ($\gamma_0$) of the pseudo-line. This pseudo-line
is collisionally coupled with the 60-GHz band (Tretyakov and Zibarova, 2018), although the impact is likely insignificant. Note
that $S_0'$ corresponds to a different definition of line intensity which has a finite nonzero value as $\nu_0 \rightarrow 0$:

$$S_0'T = \lim_{\nu \to 0} \frac{S_0 T}{\nu_0^2} \tag{4}$$

The values and uncertainties for the oxygen parameters are either from Tretyakov et al. (2005) ( Table 5) or estimated from an
independent analysis of measurement methods (Cimini et al. (2018) and references therein). The oxygen parameters perturbed




in the sensitivity analysis are listed in Table 2 (first order expansion of the line mixing parameters is adopted, as in Tretyakov et al. (2005)).

**Table 2.** List of oxygen parameters perturbed in the sensitivity analysis.

| Symbol (units) | Parameter | Uncertainty [%] | Reference |
|---|---|---|---|
| $\nu$ (kHz) | Resonant line frequency | $1.9\times10^{-6}-3.4\times10^{-5}$ | Tretyakov et al. (2005) |
| $S$ (Hz cm$^2$) | Resonant line intensity | $1-2$ | Cimini et al. (2018) and references therein |
| $n_{\mathrm{s}}$ (unitless) | Resonant line intensity temperature dependence exponent | 0.1 | Cimini et al. (2018) and references therein |
| $E_{\mathrm{low}}$ (cm$^{-1}$) | Resonant line lower-state energy | 0.25 | Cimini et al. (2018) and references therein |
| $\gamma_{\mathrm{a}}$ (GHz bar$^{-1}$) | Resonant line air-broadening | $0.82-5$ | Cimini et al. (2018) and references therein |
| $n_{\mathrm{a}}$ (unitless) | Resonant line air-broadening temperature dependence exponent | 6.25 | Cimini et al. (2018) and references therein Koshelev et al. (2016) |
| $Y$ (bar$^{-1}$) | Resonant line mixing | $1.36-27.78$ | Cimini et al. (2018) and references therein Tretyakov et al. (2005) |
| $V$ (bar$^{-1}$) | Resonant line mixing temperature dependence | $9.85-146.46$ | Cimini et al. (2018) and references therein Tretyakov et al. (2005) |
| $r_{\mathrm{w2a}}$ (unitless) | Resonant line water-to-air broadening ratio | 4.17 | Koshelev et al. (2015) |
| $\gamma_0$ (GHz bar$^{-1}$) | Zero-frequency line pressure broadening | 8.93 | Cimini et al. (2018) and references therein |
| $S'_0$ (Hz cm$^2$ GHz$^{-2}$) | Zero-frequency line intensity | 5 | Cimini et al. (2018) and references therein |

Figures 4-5 show the $\Delta T_{\mathrm{Bi},+}$ and $\Delta T_{\mathrm{Bi},-}$ spectra corresponding to the perturbation to four oxygen line absorption parameters, $(S,S'_0,\gamma_0,\gamma_{\mathrm{a}})$, and $V,Y,\nu,E_{\mathrm{low}}$, respectively. The sensitivity analysis shows that among the model parameters in Table 2, which were perturbed by the estimated uncertainty, only the following impact the modelled upwelling 16-700 GHz $T_{\mathrm{B}}$ more than 0.1 K: two for the zero- frequency non-resonant absorption $(S'_0,\gamma_0)$, four for the line position and absorption $(\nu,S,\gamma_{\mathrm{a}},n_{\mathrm{a}})$,, and two for the line mixing $(Y,V)$. Among these, the central frequency $\nu$ will not be considered for the reasons explained in Section 3. Parameters of weak oxygen lines in the 60-GHz band are included along with the strong lines because their covariance can be analyzed by the same algorithm, without incurring additional labor (except by the computer). Therefore, 109 parameters were identified as dominant for O$_2$ absorption uncertainty and are further considered for evaluation of their covariance in Section 4.





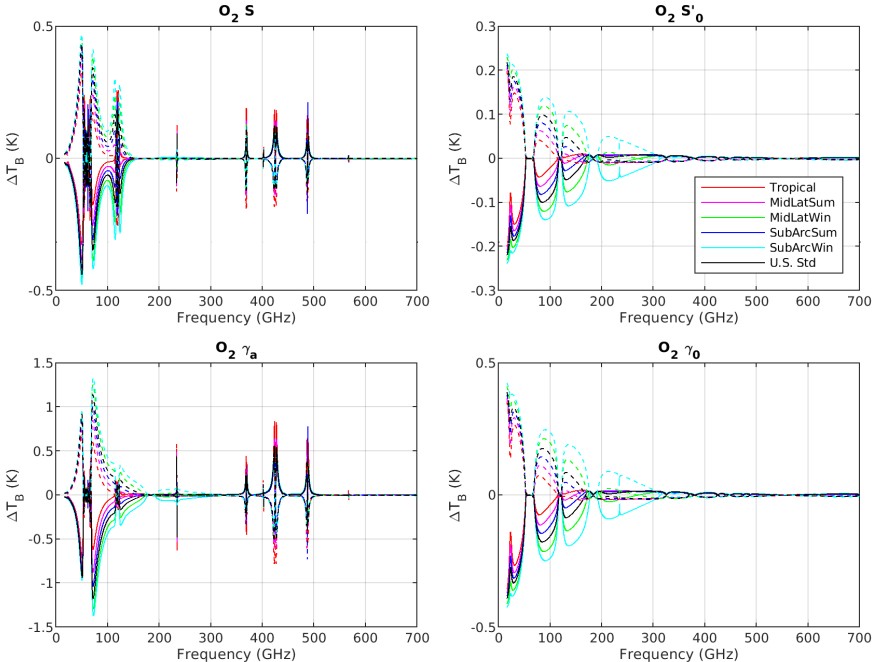

**Figure 4.** Sensitivity of modelled $T_{\mathrm{B}}$ to oxygen line absorption parameters. Solid lines correspond to negative perturbation ($\Delta \boldsymbol{T}_{\mathrm{B\,i,-}}$), while dashed lines to positive perturbation ($\Delta \boldsymbol{T}_{\mathrm{B\,i,+}}$). Top: line absorption intensity ($S$) and zero-frequency absorption intensity ($S_0'$). Bottom: line broadening ($\gamma_{\mathrm{a}}$) and zero-frequency broadening ($\gamma_0$).

## 2.3 Sensitivity to $O_3$ parameters

Ozone contributes with many lines to the absorption in the frequency range under consideration. The PWR19 model includes

the strongest 321 $O_3$ absorption lines from 100 to 800 GHz. Only resonant absorption is relevant, with the following parameters: line frequency ($\nu$), intensity ($S$), the lower-state energy ($E_{\mathrm{low}}$), air-broadening ($\gamma_{\mathrm{a}}$) and its temperature-dependence exponent ($n_{\mathrm{a}}$). The values and uncertainties are from the HITRAN 2016 database (Gordon et al. (2017)). Note that, as discussed in (Turner et al., 2022), the $O_3$ line intensity values HITRAN 2016 have been found to be 4% mis-scaled, and later adjusted in HITRAN 2020. The list of ozone parameters perturbed in the sensitivity analysis are listed in Table 3. Figure 6

shows the $\Delta \boldsymbol{T}_{\mathrm{B\,i,+}}$ and $\Delta \boldsymbol{T}_{\mathrm{B\,i,-}}$ spectra corresponding to the perturbation to ozone line absorption parameters, respectively ($\nu, S, E_{\mathrm{low}}, \gamma_{\mathrm{a}}, n_{\mathrm{a}}$). The sensitivity analysis shows that among the model parameters in Table 3, which were perturbed by the estimated uncertainty, all of these impact the modelled upwelling 16-700 GHz $\boldsymbol{T}_{\mathrm{B}}$ by more than 0.1 K.

However, we should emphasize that even though individual ozone lines contribute to the uncertainty by more than the chosen threshold, the contribution is rather small when averaged over finite channel bandwidths, due to very narrow spectral

line widths, as it will be clarified in the next Section. As a result, the ozone line parameters were not considered for evaluation of their covariance.



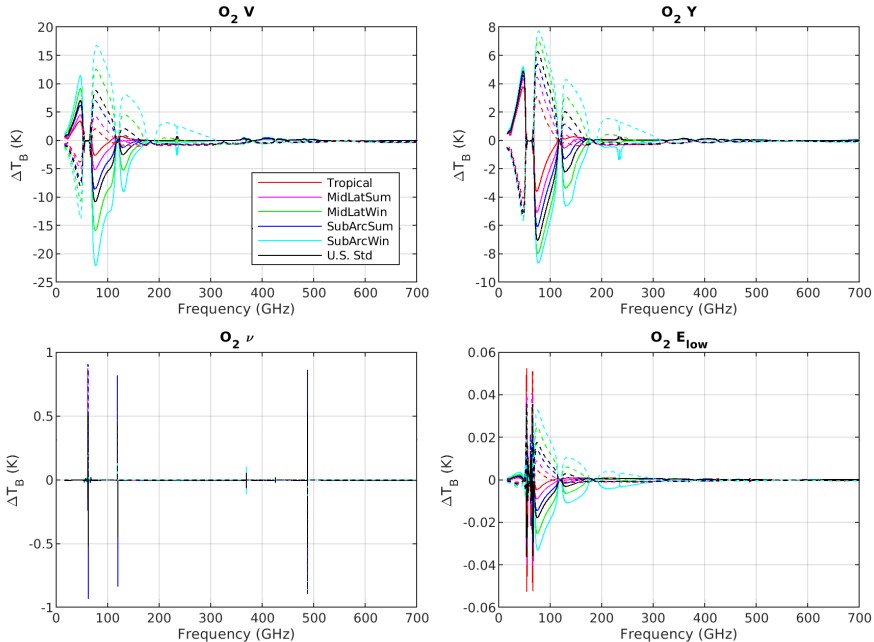

**Figure 5.** As in Fig. 4 but for line mixing ($Y$) and its temperature dependence ($V$), the central frequency ($\nu$), and the lower-state energy ($E_{\mathrm{low}}$).

**Table 3.** List of ozone parameters perturbed in the sensitivity analysis.

| Symbol (units) | Parameter | Uncertainty [%] | Reference |
|---|---|---|---|
| $\nu$ (kHz) | Resonant line frequency | $7 \times 10^{-7} - 1.8 \times 10^{-3}$ | HITRAN 2016 and references therein |
| $S$ ($\mathrm{Hz\,cm^2}$) | Resonant line intensity | 4 | HITRAN 2016 and references therein |
| $E_{\mathrm{low}}$ ($\mathrm{cm^{-1}}$) | Resonant line lower-state energy | 10 | HITRAN 2016 and references therein |
| $\gamma_{\mathrm{a}}$ ($\mathrm{GHz\,bar^{-1}}$) | Resonant line air-broadening | $5 - 20$ | HITRAN 2016 and references therein |
| $n_{\mathrm{a}}$ (unitless) | Resonant line air-broadening temperature dependence exponent | 10 | HITRAN 2016 and references therein |

## 3   Channel Convolution

The ultimate goal of this study is to characterise the absorption model uncertainty on simulated observations for selected satellite and airborne instruments, such as MWI, ICI, MWS, ATMS, MARSS, and ISMAR. Therefore, the spectral simulations,



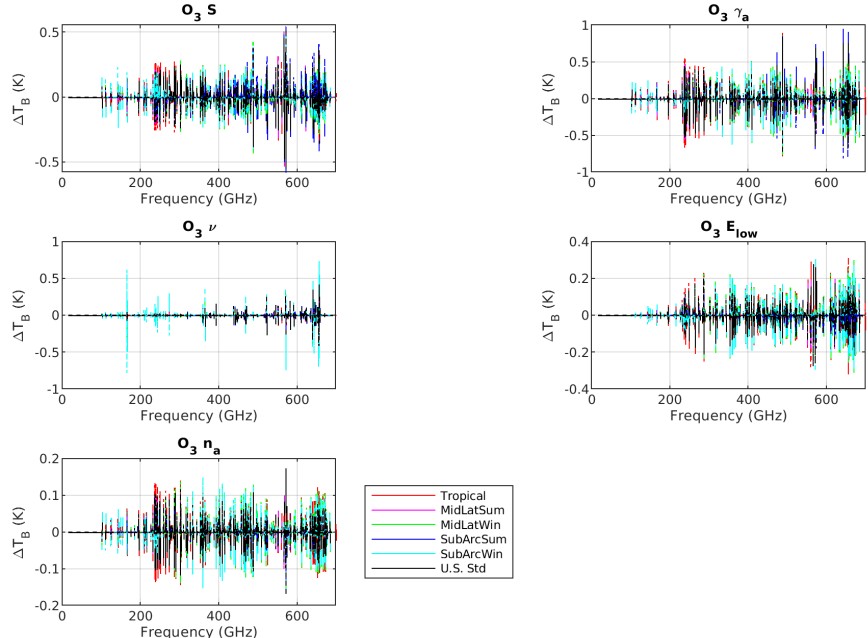

**Figure 6.** Sensitivity of modelled $T_B$ to ozone line absorption and broadening parameters. Solid lines correspond to negative perturbation ($\Delta T_{Bi,-}$), while dashed lines to positive perturbation ($\Delta T_{Bi,+}$). Top: line absorption intensity ($S$) and Air- induced broadening $\gamma_a$ coefficients. Middle: line frequency ($\nu$) and lower-state energy ($E_{low}$). Bottom: Temperature-dependence exponents of air-induced broadening ($n_a$).

as well as the associated uncertainty, need to be convolved with the channel spectral response function, which is asumed to be a simple top-hat function in this context. Table 4 (Table 5) reports the list of MWI and ICI (MWS and ATMS) channels, with their associated characteristics. Hence, before proceeding with the covariance analysis of the identified dominant parameters, a further screening is performed to discard the parameters leading to perturbations that would give negligible contribution when convolved with channel spectral response function. In particular, spectrally narrow perturbations (delta-like) are likely

to result in a negligible contribution when averaged within a channel bandwidth. As mentioned above, to simulate the channel convolution, here we consider a first-order approximation, i.e., a box-average of the simulations falling within the channel bandwidth. Considering the spectral resolution used for the calculations (50 MHz, in addition to channels' central frequencies), the number of points falling within the bandwidths in Table 4 varies, reaching up to 100 for the largest bandwidth.

     As seen in previous sections, delta-like spectrally narrow perturbations are associated with uncertainty in $H_2O$ and $O_2$

central frequency $\nu$, and all the five parameters considered for $O_3$ ($\nu, S, E_{low}, \gamma_a, n_a$). The uncertainty in central frequency $\nu$ effectively locates the absorption peak within a narrow spectral range ($\sim$100 kHz) around the absorption lines, leading to a very localised impulse going symmetrically from positive to negative. It shall be noted that most channels in Table 4 have passbands at least 100 MHz away from any line center, this being true for all channels with a bandwidth greater than the





**Table 4.** List of MWI and ICI channels with the corresponding bandwidth

| | MWI | | | ICI | | |
|---|---|---|---|---|---|---|
| Channel | Frequency (GHz) | Bandwidth (MHz) | Polarisation | Frequency (GHz) | Bandwidth (MHz) | Polarisation |
| 1 | 18.7 | 200 | H,V | 183.31±7.0 | 2×2000 | V |
| 2 | 23.8 | 400 | H,V | 183.31±3.4 | 2×1500 | V |
| 3 | 31.4 | 200 | H,V | 183.31±2.0 | 2×1500 | V |
| 4 | 50.3 | 180 | H,V | 243±2.5 | 2×3000 | H,V |
| 5 | 52.8 | 400 | H,V | 325.15±9.5 | 2×3000 | V |
| 6 | 53.24 | 400 | H,V | 325.15±3.5 | 2×2400 | V |
| 7 | 53.75 | 400 | H,V | 325.15±1.5 | 2×1600 | V |
| 8 | 89 | 4000 | H,V | 448±7.2 | 2×3000 | V |
| 9 | 118.75±3.2 | 2×500 | V | 448±3.0 | 2×2000 | V |
| 10 | 118.75±2.1 | 2×400 | V | 448±1.4 | 2×1200 | V |
| 11 | 118.75±1.4 | 2×400 | V | 664±4.2 | 2×5000 | H,V |
| 12 | 118.75±1.2 | 2×400 | V | | | |
| 13 | 165.5±0.725 | 2×3150 | V | | | |
| 14 | 183.31±7 | 2×2000 | V | | | |
| 15 | 183.31±6.1 | 2×1500 | V | | | |
| 16 | 183.31±4.9 | 2×1500 | V | | | |
| 17 | 183.31±3.4 | 2×1500 | V | | | |
| 18 | 183.31±2 | 2×1500 | V | | | |

Note that MWI channel 5 frequency has been later changed to 52.7 GHz (with 180 MHz bandwidth) to avoid issues with radio frequency interference.

detuning of the center frequency of the channel from the center of the line (i.e. 325.15±1.5 and 664±4.2 GHz channels). In
any case, although the perturbation can be large (of the order of 1 K or more), the average within a larger band would result
in negligible contribution. For these reasons, the uncertainty in central frequency $\nu$ is not further considered in the covariance
analysis in Section 4. Similarly, it can be demonstrated that the perturbations related to the uncertainty of the five parameters
considered for $O_3$ ($\nu, S, E_{\mathrm{low}}, \gamma_{\mathrm{a}}, n_{\mathrm{a}}$) have negligible effect on the band-averaged simulations, i.e. less than 60 mK for any of
the $O_3$ parameters.

**4 Estimation of uncertainty covariance matrix**

The sensitivity analysis from previous sections shows that the absorption model uncertainty on simulated upwelling $T_{\mathrm{B}}$ at finite-
bandwidth channels is dominated by the uncertainty of 26 spectroscopic parameters for water vapor and 109 parameters for
oxygen. For these parameters, the full covariance matrix of parameter uncertainties, including the off-diagonal terms giving the



**Table 5.** List of MWS and ATMS channels with the corresponding bandwidth

| | | MWS | | | ATMS | |
| --- | --- | --- | --- | --- | --- | --- |
| Channel | Frequency (GHz) | Bandwidth (MHz) | Pol. | Frequency (GHz) | Bandwidth (MHz) | Pol. |
| 1 | 23.80 | 270 | QH | 23.80 | 270 | QV |
| 2 | 31.40 | 180 | QH | 31.40 | 180 | QV |
| 3 | 50.30 | 180 | QH | 50.30 | 180 | QH |
| 4 | 52.80 | 400 | QH | 51.76 | 400 | QH |
| 5 | 53.246±0.08 | 2×140 | QH | 52.80 | 400 | QH |
| 6 | 53.596±0.115 | 2×170 | QH | 53.596±0.115 | 2×170 | QH |
| 7 | 53.948±0.081 | 2×142 | QH | 54.40 | 400 | QH |
| 8 | 54.4 | 400 | QH | 54.940 | 400 | QH |
| 9 | 54.94 | 400 | QH | 55.500 | 330 | QH |
| 10 | 55.5 | 330 | QH | 57.290344 | 330 | QH |
| 11 | 57.290344 | 330 | QH | 57.290344±0.217 | 2×78 | QH |
| 12 | 57.290344±0.217 | 2×78 | QH | 57.290344±0.3222±0.048 | 4×36 | QH |
| 13 | 57.290344±0.3222±0.048 | 4×36 | QH | 57.290344±0.3222±0.022 | 4×16 | QH |
| 14 | 57.290344±0.3222±0.022 | 4×16 | QH | 57.290344±0.3222±0.010 | 4×8 | QH |
| 15 | 57.290344±0.3222±0.010 | 4×8 | QH | 57.290344±0.3222±0.0045 | 4×3 | QH |
| 16 | 57.290344±0.3222±0.0045 | 4×3 | QH | 88.2 | 2000 | QV |
| 17 | 89 | 4000 | QV | 165.5 | 3000 | QH |
| 18 | 164-167 | 2×1350 | QH | 183.31±7.0 | 2×2000 | QH |
| 19 | 183.31±7.0 | 2×2000 | QV | 183.31±4.5 | 2×2000 | QH |
| 20 | 183.31±4.5 | 2×2000 | QV | 183.31±3.0 | 2×1000 | QH |
| 21 | 183.31±3.0 | 2×1000 | QV | 183.31±1.8 | 2×1000 | QH |
| 22 | 183.31±1.8 | 2×1000 | QV | 183.31±1.0 | 2×500 | QH |
| 23 | 183.31±1.0 | 2×500 | QV | | | |
| 24 | 229 | 2000 | QV | | | |

covariance of each parameter with the others, is required to compute the uncertainty of calculated $T_B$ at any given frequency.
The framework used to estimate the parameter covariance is described in (Rosenkranz et al., 2018; Cimini et al., 2018).
Different methods are used to estimate covariance depending on how the parameter values were measured, but some general
principles apply, as recapped hereafter. If a set of variables $a_i$ have a causal dependence on another set of variables $b_k$

$$\Delta a_i = \sum_k (\partial a_i / \partial b_k) \Delta b_k \qquad (5)$$





and the $b$ have an uncertainty covariance matrix $Cov(b)$, then

$$Cov(a_\mathrm{i}, b_\mathrm{m}) = \sum_\mathrm{k} (\partial a_\mathrm{i}/\partial b_\mathrm{k}) Cov(b_\mathrm{k}, b_\mathrm{m}) \qquad (6)$$

and the $b$ contribute to the uncertainty covariance of the $a$ the amount

$$\Delta Cov(a_\mathrm{i}, a_\mathrm{j}) = \sum_\mathrm{m} (\partial a_\mathrm{j}/\partial b_\mathrm{m}) Cov(a_\mathrm{i}, b_\mathrm{m}) \qquad (7)$$

This general principle has been used to estimate the covariance between the selected 135 parameters. Further details are given in the following subsections.

## 4.1 Covariance of $H_2O$ parameters

As described in Section 2.1, 26 parameters were identified as dominant for $H_2O$ absorption uncertainty, including continuum $(C_\mathrm{s}, C_\mathrm{f}, n_{C_\mathrm{s}}, n_{C_\mathrm{f}})$, and line $(\gamma_\mathrm{a}, n_\mathrm{a}, S)$ parameters. In fact, water vapor contributes to absorption with several resonant lines and a non-resonant absorption, the latter being by definition the remainder after the contribution of local resonant lines has been subtracted from measured absorption. Therefore, if a line parameter is revised, the continuum should also be revised to compensate and reproduce as well as possible the original measurements from which the continuum was derived. Thus, line and continuum parameters are correlated. In addition, the self and foreign continuum components are correlated, and so are their corresponding temperature-dependence exponents. The way covariance between $C_\mathrm{s}$, $C_\mathrm{f}$, $n_{C_\mathrm{s}}$ and $n_{C_\mathrm{f}}$ was derived is described in Cimini et al. (2018). The covariance between line intensities and continuum coefficients was derived in Rosenkranz et al. (2018), and it is here extended to higher frequency lines (up to 916 GHz). The covariance between air-induced line widths and continuum coefficients was derived in Cimini et al. (2018), and it is here extended to the lines for which $\gamma_\mathrm{a}$ uncertainties were found to be relevant (i.e., the six $H_2O$ key lines plus the 620 GHz line). The same approach was used to derive the covariance of line width temperature-dependence exponents with $C_\mathrm{s}$ and $C_\mathrm{f}$.

On the other hand, if two parameters are derived independently, such as one by measurement and the other from theory, or by independent measurements, then we consider them uncorrelated. Thus (using the symbols defined in Table 1), $Cov(S, n_{C_\mathrm{s}})$, $Cov(S, n_{C_\mathrm{f}})$, $Cov(S, \gamma_\mathrm{a})$, $Cov(S, n_\mathrm{a})$, $Cov(\gamma_\mathrm{a}, n_{C_\mathrm{s}})$, $Cov(\gamma_\mathrm{a}, n_{C_\mathrm{f}})$, $Cov(n_\mathrm{a}, n_{C_\mathrm{s}})$, $Cov(n_\mathrm{a}, n_{C_\mathrm{f}})$, and (with one exception, which is discussed below) $Cov(n_\mathrm{a}, \gamma_\mathrm{a})$ are all set to zero.

The intensities of different absorption lines may be slightly correlated ($\sim 0.1\%$) because of common assumptions in their theoretical calculations, but random deviation dominates ($\sim 1\%$) Conway et al. (2018), and thus we set $Cov(S_\mathrm{i}, S_\mathrm{j}) = 0$ for $\mathrm{i} \neq \mathrm{j}$.

The widths of different absorption lines and their temperature-dependence exponents were measured at low pressures such that they do not overlap, and thus have independent uncertainties. Thus, $Cov(\gamma_{\mathrm{a,i}}, \gamma_{\mathrm{a,j}}) = Cov(n_{\mathrm{a,i}}, n_{\mathrm{a,j}}) = 0$ for $\mathrm{i} \neq \mathrm{j}$.

The exception noted above is at 325 GHz, where $n_\mathrm{a}$ comes from the measurements by Colmont et al. (1999), who derived $\gamma_\mathrm{a}$ and $n_\mathrm{a}$ as the intercept and slope of a linear fit between $ln(\gamma_\mathrm{a})$ and $ln(T)$. As such, the model to be fitted

$$\gamma_\mathrm{a}(T) = \gamma_\mathrm{a}(T_0)(T_0/T)^{n_\mathrm{a}} \qquad (8)$$



can be written as

$$y = a + bx \tag{9}$$

where

$$y = ln(\gamma_{\mathrm{a}}) \qquad a = ln(\gamma_{\mathrm{a}}(T_0)) \qquad b = n_{\mathrm{a}} \qquad x = ln(T_0/T) \tag{10}$$

The two parameters $a$ and $b$ are simultaneously fitted by least-squares, so the reasoning in Cimini et al. (2018) (Section

4.1.3) applies. If the uncertainty in $\gamma_a$ is small compared to its value, then the correlation between $\gamma_a$ and $n_{\mathrm{a}}$ is

$$\rho(\gamma_{\mathrm{a}}, n_{\mathrm{a}}) = \rho(a,b) = - <x> / \sqrt{\sigma_x^2 + <x>^2} \tag{11}$$

where $<x>$ is the average value of $x$ and $\sigma_x$ is its standard deviation.

### 4.2   Covariance of $O_2$ parameters

As described in Section 2.2, 109 parameters were identified as dominant for $O_2$ absorption uncertainty, including zero-

frequency continuum ($S'_0, \gamma_0$), line shape ($S, \gamma_{\mathrm{a}}, n_{\mathrm{a}}$) and line mixing ($Y, V$) parameters. Concerning the continuum absorption, it is very difficult to measure the broadening ($\gamma_0$) independently of the intensity ($S'_0$) for this zero-frequency pseudo-line. For that reason, Cimini et al. (2018) suggested that only the uncertainty of $\gamma_0$ could be used as a surrogate for the combination of $S'_0$ and $\gamma_0$. The estimated uncertainty is based on the spread of published measurements, accounting for the combination of intensity and broadening uncertainties. Concerning the line mixing ($Y, V$), only parameters for the first 34 lines (quantum

number N $< 33 + /-$) are included. The neglected lines, 15 in total, correspond to the 4 weakest lines of the 60 GHz band (50.9877, 68.4310, 50.4742, 68.9603 GHz), which are at least one order of magnitude weaker than the others, and 9 sub-mm lines, which are not significantly affected by line mixing at atmospheric pressures. Their contribution has been evaluated as negligible up to 20% uncertainty in $Y$ and $V$.

The covariance between the other parameters has been evaluated as in Cimini et al. (2018), with the following exceptions:

In addition to the 34 lines above (quantum number N $< 33 + /-$), also the air-induced line widths ($\gamma_{\mathrm{a}}$) of 4 sub-mm lines are considered relevant, i.e., 234, 368, 424, and 487 GHz. These are assumed as uncorrelated to other parameters, as they are not affected by mixing and have been derived independently from intensity.

### 5   Uncertainty propagation to brightness temperatures simulations

In the previous section the full uncertainty covariance matrix $Cov(p)$ has been estimated for the set of 135 dominant parameters

for water vapor and oxygen absorption. Thus, the full uncertainty covariance matrix for the computed $T_{\mathrm{B}}$ can be derived from Eq. 2, where $K_p$ is the Jacobian of the radiative transfer model with respect to the spectroscopic parameters p, which is computed by small perturbation analysis. The next two sections present the results of the radiative transfer simulations, focusing firstly (Sec. 5.1) on the upwelling $T_{\mathrm{B}}$, as seen from satellite sensors. Then Section 5.2 applies the above framework to upward looking geometry.





## 5.1 Downlooking view

The full $T_{\text{B}}$ uncertainty covariance matrix corresponding to the lump contribution of the 135 dominant $H_2O$ and $O_2$ spectroscopic parameters has been computed from Eq. 2 for the six typical climatology conditions introduced earlier (Anderson et al. (1986)). Figure 7 shows the square root of the diagonal terms of each of the six $T_{\text{B}}$ uncertainty covariance matrices, i.e., the $T_{\text{B}}$ uncertainty spectra due to the 135 dominant $H_2O$ and $O_2$ spectroscopic parameters for each of the six climatology conditions.

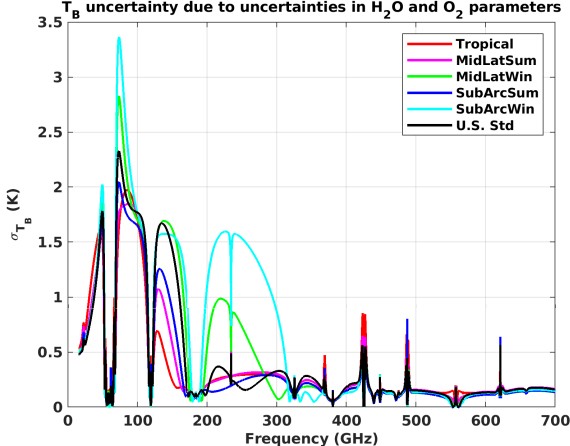

**Figure 7.** Brightness temperature uncertainty for downward looking view at $53°$, due to uncertainties in $H_2O$ and $O_2$ parameters. Six typical climatology conditions are considered (tropical, midlatitude summer, midlatitude winter, sub-arctic summer, sub-arctic winter, U.S. standard).

We notice that uncertainty in the mm-wave range is dominated by the water vapor continuum and the oxygen line mixing, with uncertainties in brightness temperature that reach up to 3.5K. Conversely, in the sub-mm range the uncertainty due to water vapor absorption lines dominates over the continuum absorption, as higher frequency lines are very opaque and thus even the wings are stronger than the continuum absorption, which is relatively weaker in the mid to upper atmosphere, due to the quadratic dependence on water vapour pressure.

As previously mentioned, the major goal of this work is to provide uncertainties on synthetic $T_{\text{B}}$, relative to channels of EPS-SG imagers and sounders. Hence we have performed the convolution of the spectra in Fig. 7 with top-hat functions corresponding to channel bandwidth reported in Table 4 and 5 to estimate the corresponding uncertainty on simulated $T_{\text{B}}$ for MWI, ICI, MWS and ATMS. The results are shown in Figure 8, with the uncertainty on simulated $T_{\text{B}}$ computed for the six considered climatology conditions. It can be seen that generally the estimated uncertainty is not negligible, and can reach more than 2 K. For ICI, the estimated uncertainty is quite small, less than 0.2 K, for all channels except channel 4, which has large uncertainty in cold and dry environments. MWS and ATMS sounders show very similar features, since they mostly have the same frequency channels: only a couple of channels show an uncertainty larger than 1 K, while the others feature smaller, though non-negligible, uncertainty values. These differences stem from different contributions from line and



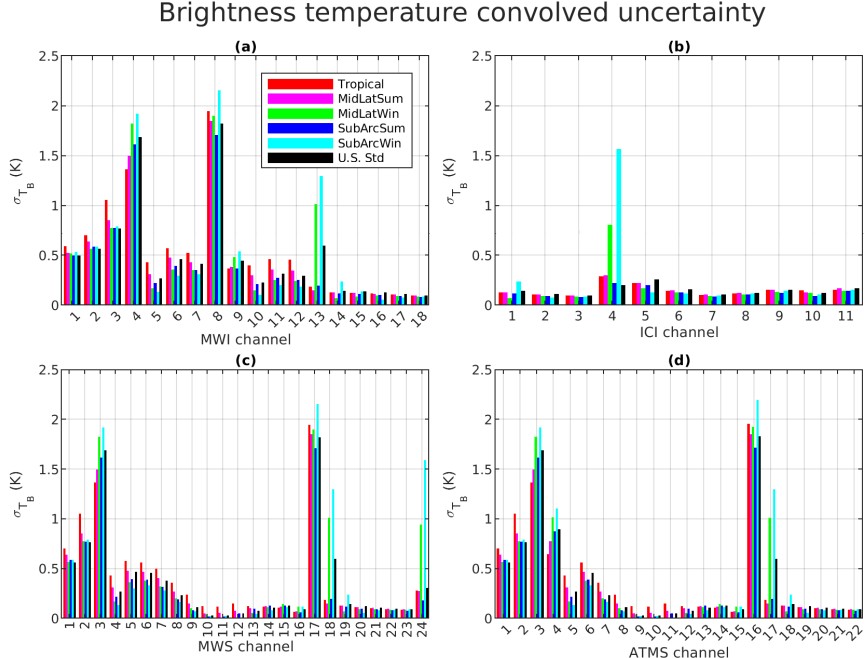

**Figure 8.** Brightness temperature uncertainty convolved on MWI and ICI (top), MWS and ATMS (bottom) channels.

continuum absorption. This confirms that the uncertainty of brightness temperature simulations cannot be assumed negligible
when comparing simulations with observations, such as within satellite sensor calibration/validation efforts, as they are of the
same order or even larger than typical radiometric accuracy ($0.7 - 2.0\,\mathrm{K}$ for MWI/ICI).

## 5.2 Upward looking view

While previous sections considers down-looking from top-of-the-atmosphere (TOA), this section investigates the uncertainty
associated with down-welling $T_\mathrm{B}$, relative to the upward looking geometry feasible with airborne sensors (e.g., Fox et al.
(2023)). Note that the same covariance matrix for 135 parameters is assumed here, while more rigorously the sensitivity should
be reevaluated at each height. However, the 111 parameters that were selected in the previous study by Cimini et al. (2018)
are included, although they were identified as dominant in the spectral range limited to $20 - 150\,\mathrm{GHz}$. The six climatology
conditions described earlier are used to compute the uncertainty covariance matrix corresponding to the lump contribution of
the 135 dominant $H_2O$ and $O_2$ spectroscopic parameters. But considering that airborne sensors typically change their altitude
during the flight, the full $T_\mathrm{B}$ uncertainty covariance matrix has been computed assuming upward-looking zenith views from a
set of 9 altitudes (i.e., 0, 1, 2, 3, 5, 7, 8, 9, 10 km).

Figure 9 shows the square root of the diagonal terms of each resulting $T_\mathrm{B}$ uncertainty covariance matrix, i.e., the $T_\mathrm{B}$ uncertainty spectra due to the 135 dominant $H_2O$ and $O_2$ spectroscopic parameters for each of the six climatology conditions.





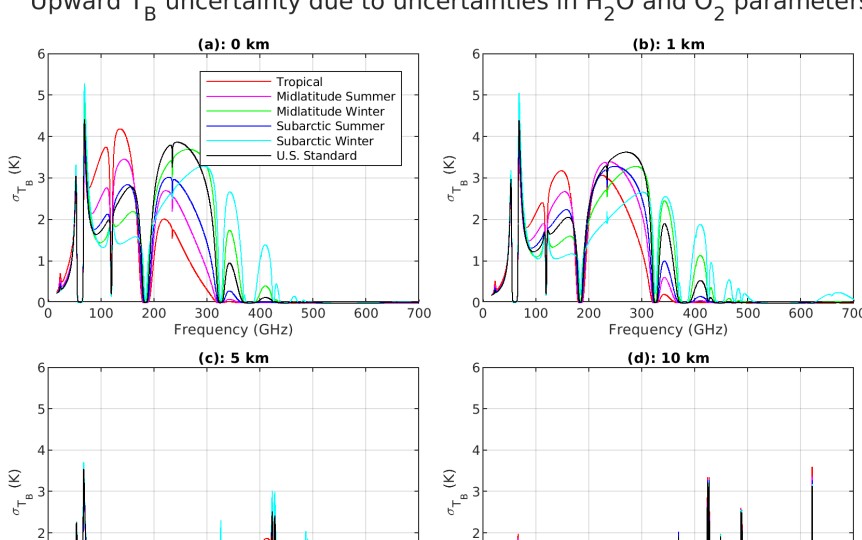

**Figure 9.** Brightness temperature uncertainty for upward looking view, due to uncertainties in $H_2O$ and $O_2$ parameters. Top: from $0\,\mathrm{km}$ (left) and $1\,\mathrm{km}$ (right) height; Bottom: $5\,\mathrm{km}$ (left) and $10\,\mathrm{km}$ (right) height. Six typical climatology conditions are considered (tropical, midlatitude summer, midlatitude winter, sub-arctic summer, sub-arctic winter, U.S. standard).

Four altitudes are shown, representative of observations from near surface ($0\,\mathrm{km}$), within the boundary layer ($1\,\mathrm{km}$), free tro-
350 posphere ($5\,\mathrm{km}$), and high troposphere ($10\,\mathrm{km}$). It can be seen that at low altitudes the uncertainty is dominated by water vapor continuum uncertainties (except for the $60\,\mathrm{GHz}$ band), while at high altitudes the uncertainty is dominated by line absorption uncertainties, due to the different pressure-dependence of continuum (quadratic) and line (linear) absorption.

As done in Sec. 5.1 for the down-looking geometry, the spectra in Figure 9 can be convolved with the top-hat function cor-
responding to channel bandwidths in Table 6 to estimate the corresponding uncertainty on simulated downwelling $T_{\mathrm{B}}$. This
is performed only for ISMAR and MARSS channels, i.e. the two airborne instruments demonstrator considered in this study, since the upward-looking view from spaceborne sensors at TOA does not encounter the Earth's atmosphere. Accordingly, Figure 10-11 show the uncertainty on simulated $T_{\mathrm{B}}$ for ISMAR and MARSS channels computed for the six considered climatology conditions from four representative altitudes: near surface ($0\,\mathrm{km}$), within the boundary layer ($1\,\mathrm{km}$), free troposphere ($5\,\mathrm{km}$), and high troposphere ($10\,\mathrm{km}$). The resulting estimated uncertainty has been used in a companion paper to constrain
observation minus simulation statistics collected in several airborne campaigns deploying ISMAR and MARSS (Fox et al. (2023)).



**Table 6.** List of ISMAR and MARSS channels with the corresponding bandwidth

| | ISMAR | | MARSS | |
| Channel | Frequency (GHz) | Bandwidth (MHz) | Frequency (GHz) | Bandwidth (MHz) |
| --- | --- | --- | --- | --- |
| 1 | $118.75 \pm 1.1$ | $2 \times 400$ | $88.99 \pm 1.1$ | $2 \times 650$ |
| 2 | $118.75 \pm 1.5$ | $2 \times 400$ | $157.075 \pm 2.6$ | $2 \times 2600$ |
| 3 | $118.75 \pm 2.1$ | $2 \times 800$ | $183.248 \pm 0.975$ | $2 \times 450$ |
| 4 | $118.75 \pm 3.0$ | $2 \times 1000$ | $183.248 \pm 3$ | $2 \times 1000$ |
| 5 | $118.75 \pm 5.0$ | $2 \times 2000$ | $183.248 \pm 7$ | $2 \times 2000$ |
| 6 | $243.2 \pm 2.5$ | $2 \times 3000$ | | |
| 7 | $325.15 \pm 1.5$ | $2 \times 1600$ | | |
| 8 | $325.15 \pm 3.5$ | $2 \times 2400$ | | |
| 9 | $325.15 \pm 9.5$ | $2 \times 3000$ | | |
| 10 | $424.7 \pm 1$ | $2 \times 400$ | | |
| 11 | $424.7 \pm 1.5$ | $2 \times 600$ | | |
| 12 | $424.7 \pm 4$ | $2 \times 1000$ | | |
| 13 | $448 \pm 1.4$ | $2 \times 1200$ | | |
| 14 | $448 \pm 3$ | $2 \times 2000$ | | |
| 15 | $448 \pm 7.2$ | $2 \times 3000$ | | |
| 16 | $664 \pm 4.2$ | $2 \times 5000$ | | |
| 17 | $874.4 \pm 6$ | $2 \times 4040$ | | |

## 6 Summary and conclusions

This paper quantifies the uncertainty of microwave radiative transfer calculations due to uncertainty of spectroscopic parameters in the framework of preparatory activity for EPS-SG microwave radiometer. First, the sensitivity of radiative transfer
calculations in the microwave to millimeter-wave range has been evaluated against the uncertainty of spectroscopic parameters for $H_2O$, $O_2$, and $O_3$, adopting the observing geometry typical of satellite imagers such as MWI and ICI (down-looking from TOA at $53°$ incident angle) and surface emissivity for typical ocean conditions at H-polarization, which is more conservative than V-pol for estimating the uncertainty related to the atmospheric absorption model. Note that uncertainties at $53°$ incident angle could be assumed as the higher boundary for cross scanning instruments (such as MWS and ATMS), as the atmospheric
path gets shorter at higher incident angles.

The sensitivity analysis identified a set of 135 spectroscopic parameters as dominant for the uncertainty of simulated $T_B$ (26 for $H_2O$ and 109 for $O_2$) while $O_3$ was judged to contribute only negligibly to the uncertainty of finite-bandwidth channels. The full uncertainty covariance matrix for the 135 spectroscopic parameters has been evaluated, including the off-diagonal terms indicating the cross-covariance between parameter uncertainties. Thus, the full $T_B$ uncertainty covariance matrix, cor-
375 responding to the lump contribution of the 135 dominant $H_2O$ and $O_2$ spectroscopic parameters, has been computed for six

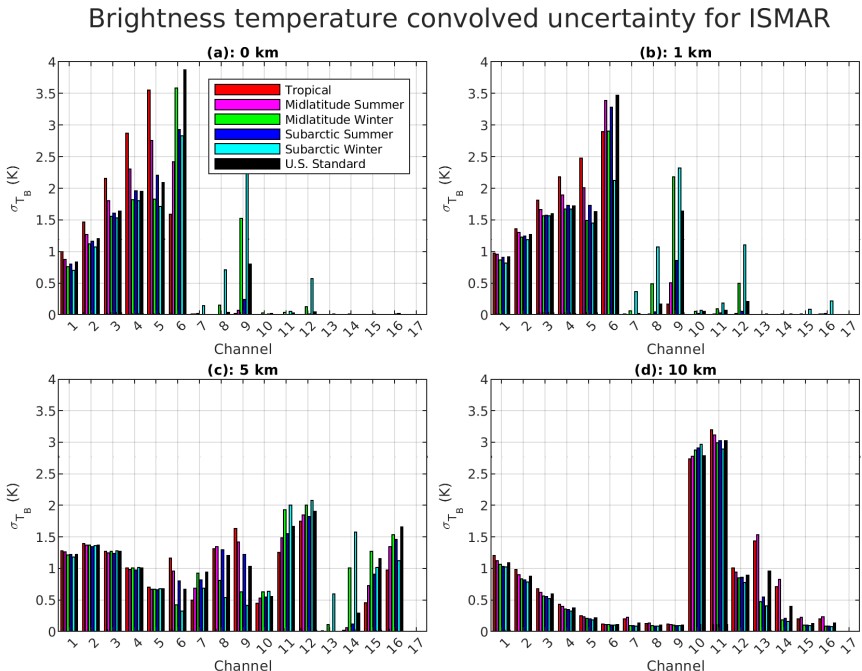

**Figure 10.** Brightness Temperature uncertainty convolved on ISMAR channels (as in Table 6). ISMAR channel 17 is not shown as it lies outside of the frequency range considered in this study. Top: 0 (left) and 1 km (right) height; Bottom: 5 (left) and 10 km (right) height

climatology conditions (tropical, midlatitude summer, midlatitude winter, sub-arctic summer, sub-arctic winter, U.S. standard). Finally, the $T_{\mathrm{B}}$ uncertainty spectrum has been computed for each of the six different climatology conditions, as the square root of the diagonal terms of $T_{\mathrm{B}}$ uncertainty covariance matrices. The uncertainty of simulated $T_{\mathrm{B}}$ has been also evaluated for MWI, ICI, MWS and ATMS channels, considering their nominal bandpass filters, ranging from 0.1 K at relatively opaque channels

to 2.2 K at relatively transparent channels (all numerical values are reported in Table A1, Table A2, Table A3, Table A4). These uncertainties are strictly valid over ocean surface (covering 72 % of the globe), and are deemed conservative with respect to other surface background, which usually have higher emissivity than the ocean. For example, the channel uncertainty has been evaluated using typical sea-ice emissivity, showing lower values throughout the spectral domain, and especially at lower frequency (10-100 GHz) for which sea-ice emissivity gets closer to one. The channel uncertainty was also quantified for two

airborne instruments (ISMAR and MARSS) assuming zenith upward-looking observations at different aircraft altitudes (0-10 km), showing values from just above 0.0 to 3.8 K, depending on channel opacity and assumed climatology.

The analysis above was obtained using PWR19. Therefore, the quantified uncertainties are strictly valid for this model. The uncertainty of other absorption models, adopting different spectroscopy, could be evaluated with the same approach. One relevant absorption model is that developed and maintained by AER inc. (Clough et al. (2005); Cady-Pereira et al. (2020)) adopting

the MT-CKD water vapor continuum model (Mlawer et al. (2019, 2012)), as it was used to train the ICI coefficients for the fast RTM adopted for the ICI operational retrievals and data assimilation in NWP models (RTTOV version 13). Considerations on



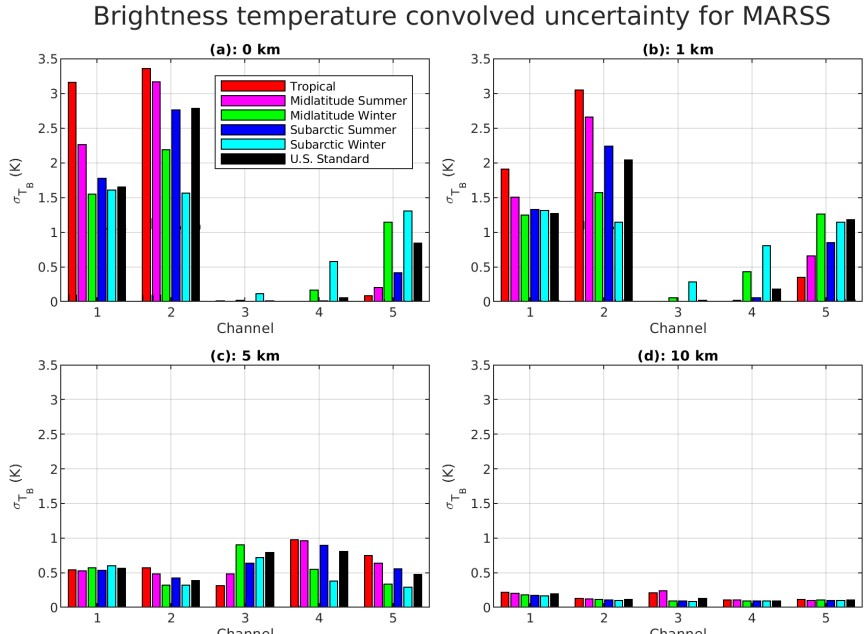

**Figure 11.** As in Fig. 10 but for MARSS channels in Table 6

the characteristics of the AER and MT-CKD model, with respect to PWR19, indicate that uncertainty in the $H_2O$ continuum would decrease by half due to smaller continuum coefficients' uncertainty (by roughly 50%, Mlawer, personal communication 2021, although the difference is more complex as highlighted by Odintsova et al. (2017)), while the uncertainty deviation due to $H_2O$ line absorption would be small (<0.1 K) except in the 20-25 GHz range ($\sim 0.8$K increase). . The uncertainty due to $O_2$ parameters is expected to be the same, as PWR and AER models share the same $O_2$ spectroscopic parameters from Tretyakov et al. (2005). However, such a speculative analysis is limited by the fact that the MT-CKD formulation is more complicated - i.e., the parameters vary with frequency/wavenumber - and that we were not able to find information concerning the correlation between MT-CKD continuum coefficients and the way $O_2$ line mixing parameters and their temperature dependence are used within the AER code.

Finally, while this analysis was finalised, an updated spectroscopy was released (PWR22, available at http://cetemps.aquila. infn.it/mwrnet/lblmrt_ns.html). Appendix B reports expected systematic and random differences between PWR22 and PWR19; this gives an indication of the additional uncertainty of PWR19 with respect to the latest version, which includes more mm-wave WV lines and more recent spectroscopic findings, while an extension of the uncertainty analysis to PWR22 is planned as future work.



*Code and data availability.* Uncertainty covariance matrices for the spectroscopic parameters considered here, as well as the resulting $T_B$ uncertainty covariance matrices for all channels, are available as a supplement to this paper. The absorption model by Rosenkranz (2019), as well as newer and older versions, is available as a FORTRAN 77 code at http://cetemps.aquila.infn.it/mwrnet/lblmrt_ns.html (last access: 6 October 2023). See also Larosa et al. (2023) for a python-based code implementation.

**Appendix A: Uncertainty Values**

In this section we report the values for top-of-atmosphere upwelling brightness temperature uncertainty (at 1-sigma level) arising from the uncertainty covariance of 135 spectroscopic parameters identified as dominant (109 related to $O_2$ absorption, 26 related to $H_2O$ absorption) for channels of the MicroWave Imager (Table A1), Ice Cloud Imager (Table A2), MicroWave Sounder (Table A3) and Advanced Technology Microwave Sounder (Table A4). The convolution with a top-hat response func-
415 tion, taking into account a channel bandwidth is computed for both horizontal and vertical polarisation, for each of the six climatology atmospheric profiles.





**Table A1.** Uncertainty for simulated TOA upwelling $T_B$[K] at MWI channels [GHz] due to uncertainties in $H_2O$ and $O_2$ parameters. Six climatological atmospheric conditions are considered: tropical, midlatitude summer, midlatitude winter, sub-arctic summer, sub-arctic winter, U.S. standard.

| MWI | | | | | | |
|---|---|---|---|---|---|---|
| Channel (GHz) | Tropical | MidLatSum | MidLatWint | SubArcSum | SubArcWin | U.S. Std |
| (Polarisation) | (H V) | (H V) | (H V) | (H V) | (H V) | (H V) |
| 1) 18.7 | (0.59 0.32) | (0.52 0.28) | (0.52 0.28) | (0.50 0.27) | (0.54 0.30) | (0.50 0.27) |
| 2) 23.8 | (0.70 0.37) | (0.64 0.34) | (0.57 0.30) | (0.59 0.31) | (0.59 0.32) | (0.57 0.29) |
| 3) 31.4 | (1.06 0.55) | (0.85 0.44) | (0.78 0.40) | (0.77 0.39) | (0.79 0.41) | (0.77 0.38) |
| 4) 50.3 | (1.37 0.51) | (1.50 0.57) | (1.82 0.74) | (1.62 0.62) | (1.92 0.83) | (1.69 0.62) |
| 5) 52.8 | (0.43 0.57) | (0.31 0.47) | (0.17 0.33) | (0.22 0.39) | (0.14 0.25) | (0.27 0.47) |
| 6) 53.24 | (0.57 0.63) | (0.48 0.53) | (0.36 0.43) | (0.39 0.46) | (0.30 0.37) | (0.46 0.54) |
| 7) 53.75 | (0.52 0.53) | (0.43 0.44) | (0.35 0.36) | (0.35 0.36) | (0.31 0.32) | (0.42 0.43) |
| 8) 89 | (1.95 0.78) | (1.85 0.74) | (1.90 0.73) | (1.71 0.66) | (2.16 0.86) | (1.82 0.68) |
| 9) 118.75±3.2 | (0.37 0.31) | (0.39 0.25) | (0.48 0.12) | (0.37 0.18) | (0.54 0.11) | (0.45 0.19) |
| 10) 118.75±2.1 | (0.40 0.43) | (0.30 0.34) | (0.15 0.24) | (0.21 0.28) | (0.11 0.19) | (0.23 0.32) |
| 11) 118.75±1.4 | (0.47 0.47) | (0.36 0.37) | (0.26 0.27) | (0.27 0.29) | (0.20 0.22) | (0.32 0.34) |
| 12) 118.75±1.2 | (0.46 0.46) | (0.35 0.35) | (0.24 0.25) | (0.26 0.26) | (0.19 0.19) | (0.30 0.31) |
| 13) 165.5±0.725 | (0.19 0.20) | (0.15 0.18) | (1.01 0.29) | (0.20 0.12) | (1.30 0.43) | (0.60 0.13) |
| 14) 183.31±7 | (0.13 0.13) | (0.13 0.13) | (0.07 0.09) | (0.12 0.12) | (0.24 0.07) | (0.15 0.15) |
| 15) 183.31±6.1 | (0.12 0.12) | (0.12 0.12) | (0.09 0.10) | (0.11 0.11) | (0.14 0.05) | (0.14 0.14) |
| 16 ) 183.31±4.9 | (0.12 0.12) | (0.12 0.12) | (0.10 0.10) | (0.10 0.10) | (0.06 0.06) | (0.13 0.13) |
| 17) 183.31±3.4 | (0.11 0.11) | (0.11 0.11) | (0.09 0.09) | (0.09 0.09) | (0.08 0.08) | (0.11 0.11) |
| 18) 183.31±2 | (0.10 0.10) | (0.10 0.10) | (0.09 0.09) | (0.09 0.09) | (0.09 0.09) | (0.10 0.10) |



**Table A2.** As in Table A1, but for ICI.

| | ICI | | | | | |
|---|---|---|---|---|---|---|
| Channel (GHz) (Polarisation) | Tropical (H V) | MidLatSum (H V) | MidLatWint (H V) | SubArcSum (H V) | SubArcWin (H V) | U.S. Std (H V) |
| 1) 183.31±7.0 | (0.13 0.13) | (0.13 0.13) | (0.07 0.09) | (0.12 0.12) | (0.24 0.07) | (0.15 0.15) |
| 2) 183.31±3.4 | (0.11 0.11) | (0.11 0.11) | (0.09 0.09) | (0.09 0.09) | (0.08 0.08) | (0.11 0.11) |
| 3) 183.31±2.0 | (0.10 0.10) | (0.10 0.10) | (0.09 0.09) | (0.09 0.09) | (0.09 0.09) | (0.10 0.10) |
| 4) 243±2.5 | (0.29 0.29) | (0.30 0.31) | (0.81 0.12) | (0.22 0.28) | (1.57 0.41) | (0.20 0.26) |
| 5) 325.15±9.5 | (0.22 0.22) | (0.22 0.22) | (0.17 0.18) | (0.20 0.20) | (0.13 0.08) | (0.26 0.26) |
| 6) 325.15±3.5 | (0.15 0.15) | (0.15 0.15) | (0.13 0.13) | (0.13 0.13) | (0.12 0.12) | (0.16 0.16) |
| 7) 325.15±1.5 | (0.10 0.10) | (0.11 0.11) | (0.09 0.09) | (0.09 0.09) | (0.10 0.10) | (0.11 0.11) |
| 8) 448±7.2 | (0.12 0.12) | (0.13 0.13) | (0.11 0.11) | (0.11 0.13) | (0.12 0.12) | (0.13 0.13) |
| 9) 448±3.0 | (0.15 0.15) | (0.16 0.16) | (0.14 0.14) | (0.13 0.13) | (0.15 0.15) | (0.16 0.16) |
| 10) 448±1.4 | (0.15 0.15) | (0.13 0.13) | (0.12 0.12) | (0.10 0.10) | (0.11 0.11) | (0.13 0.13) |
| 11) 664±4.2 | (0.16 0.16) | (0.17 0.17) | (0.15 0.15) | (0.15 0.15) | (0.16 0.16) | (0.17 0.17) |





**Table A3.** As in Table A1, but for MWS.

| | MWS | | | | | |
|---|---|---|---|---|---|---|
| Channel (GHz) | Tropical | MidLatSum | MidLatWint | SubArcSum | SubArcWin | U.S. Std |
| (Polarisation) | (H V) | (H V) | (H V) | (H V) | (H V) | (H V) |
| 1) 23.8 | (0.70 0.37) | (0.64 0.34) | (0.57 0.30) | (0.59 0.31) | (0.59 0.32) | (0.56 0.29) |
| 2) 31.4 | (1.06 0.55) | (0.85 0.44) | (0.78 0.40) | (0.77 0.39) | (0.79 0.41) | (0.77 0.38) |
| 3) 50.3 | (1.37 0.51) | (1.50 0.57) | (1.82 0.74) | (1.62 0.62) | (1.92 0.83) | (1.69 0.62) |
| 4) 52.8 | (0.43 0.57) | (0.31 0.46) | (0.17 0.33) | (0.22 0.39) | (0.14 0.25) | (0.27 0.47) |
| 5) 53.246±0.08 | (0.58 0.63) | (0.48 0.53) | (0.36 0.43) | (0.40 0.46) | (0.30 0.37) | (0.47 0.54) |
| 6) 53.596±0.115 | (0.56 0.58) | (0.47 0.49) | (0.38 0.40) | (0.39 0.41) | (0.33 0.35) | (0.46 0.48) |
| 7) 53.948±0.081 | (0.50 0.50) | (0.41 0.41) | (0.32 0.33) | (0.32 0.32) | (0.28 0.29) | (0.38 0.38) |
| 8) 54.4 | (0.36 0.36) | (0.27 0.27) | (0.20 0.20) | (0.19 0.19) | (0.17 0.17) | (0.23 0.24) |
| 9) 54.94 | (0.24 0.24) | (0.15 0.15) | (0.10 0.10) | (0.08 0.08) | (0.07 0.07) | (0.11 0.11) |
| 10) 55.5 | (0.12 0.12) | (0.05 0.05) | (0.04 0.04) | (0.02 0.02) | (0.03 0.03) | (0.03 0.03) |
| 11) 57.290344 | (0.12 0.12) | (0.06 0.06) | (0.01 0.01) | (0.02 0.02) | (0.02 0.02) | (0.03 0.03) |
| 12) 57.290344±0.217 | (0.15 0.15) | (0.08 0.08) | (0.01 0.01) | (0.05 0.05) | (0.01 0.01) | (0.05 0.05) |
| 13) 57.290344±0.3222±0.048 | (0.13 0.13) | (0.11 0.11) | (0.05 0.05) | (0.10 0.10) | (0.05 0.05) | (0.08 0.08) |
| 14) 57.290344±0.3222±0.022 | (0.12 0.12) | (0.12 0.12) | (0.11 0.11) | (0.13 0.13) | (0.11 0.08) | (0.08 0.11) |
| 15) 57.290344±0.3222±0.010 | (0.11 0.11) | (0.12 0.12) | (0.15 0.15) | (0.13 0.13) | (0.12 0.12) | (0.13 0.13) |
| 16) 57.290344±0.3222±0.0045 | (0.07 0.07) | (0.07 0.07) | (0.12 0.12) | (0.06 0.06) | (0.12 0.12) | (0.10 0.09) |
| 17) 89 | (1.95 0.78) | (1.85 0.74) | (1.90 0.73) | (1.71 0.66) | (2.16 0.86) | (1.82 0.68) |
| 18) 165.5±0.725 | (0.19 0.20) | (0.15 0.18) | (1.01 0.29) | (0.20 0.12) | (1.30 0.43) | (0.60 0.13) |
| 19) 183.31±7 | (0.13 0.13) | (0.13 0.13) | (0.07 0.09) | (0.12 0.12) | (0.24 0.07) | (0.15 0.15) |
| 20) 183.31±4.5 | (0.12 0.12) | (0.11 0.11) | (0.10 0.10) | (0.10 0.10) | (0.06 0.07) | (0.12 0.12) |
| 21 ) 183.31±3.0 | (0.11 0.11) | (0.11 0.11) | (0.09 0.09) | (0.09 0.09) | (0.08 0.09) | (0.11 0.11) |
| 22) 183.31±1.8 | (0.09 0.09) | (0.10 0.10) | (0.09 0.09) | (0.08 0.08) | (0.09 0.09) | (0.10 0.10) |
| 23) 183.31±1 | (0.09 0.09) | (0.09 0.09) | (0.09 0.09) | (0.08 0.08) | (0.10 0.10) | (0.09 0.09) |
| 24) 229 | (0.28 0.28) | (0.28 0.29) | (0.95 0.17) | (0.18 0.26) | (1.60 0.44) | (0.31 0.21) |



**Table A4.** As in Table A1, but for ATMS.

| | ATMS | | | | | |
|---|---|---|---|---|---|---|
| Channel (GHz) | Tropical | MidLatSum | MidLatWint | SubArcSum | SubArcWin | U.S. Std |
| (Polarisation) | (H V) | (H V) | (H V) | (H V) | (H V) | (H V) |
| 1) 23.8 | (0.70 0.37) | (0.64 0.34) | (0.57 0.30) | (0.59 0.31) | (0.59 0.32) | (0.56 0.29) |
| 2) 31.4 | (1.06 0.55) | (0.85 0.44) | (0.78 0.40) | (0.77 0.39) | (0.79 0.41) | (0.77 0.38) |
| 3) 50.3 | (1.37 0.51) | (1.50 0.57) | (1.82 0.74) | (1.62 0.62) | (1.92 0.83) | (1.69 0.62) |
| 4) 51.76 | (0.65 0.26) | (0.77 0.22) | (1.02 0.27) | (0.88 0.21) | (1.10 0.36) | (0.90 0.20) |
| 5) 52.8 | (0.43 0.57) | (0.31 0.47) | (0.17 0.33) | (0.22 0.39) | (0.14 0.25) | (0.27 0.47) |
| 6) 53.596±0.115 | (0.56 0.58) | (0.47 0.49) | (0.38 0.40) | (0.39 0.41) | (0.33 0.35) | (0.46 0.48) |
| 7) 54.4 | (0.36 0.36) | (0.27 0.27) | (0.20 0.20) | (0.19 0.19) | (0.17 0.17) | (0.23 0.24) |
| 8) 54.94 | (0.24 0.24) | (0.15 0.15) | (0.10 0.10) | (0.08 0.08) | (0.07 0.07) | (0.11 0.11) |
| 9) 55.5 | (0.12 0.12) | (0.05 0.05) | (0.04 0.04) | (0.02 0.02) | (0.03 0.03) | (0.03 0.03) |
| 10) 57.290344 | (0.12 0.12) | (0.06 0.06) | (0.01 0.01) | (0.02 0.02) | (0.02 0.02) | (0.03 0.03) |
| 11) 57.290344±0.217 | (0.15 0.15) | (0.08 0.08) | (0.01 0.01) | (0.05 0.05) | (0.01 0.01) | (0.05 0.08) |
| 12) 57.290344±0.3222±0.048 | (0.13 0.13) | (0.11 0.11) | (0.05 0.05) | (0.10 0.10) | (0.05 0.05) | (0.08 0.11) |
| 13) 57.290344±0.3222±0.022 | (0.12 0.12) | (0.12 0.12) | (0.11 0.11) | (0.13 0.13) | (0.08 0.08) | (0.11 0.13) |
| 14) 57.290344±0.3222±0.010 | (0.11 0.11) | (0.12 0.12) | (0.15 0.15) | (0.13 0.13) | (0.12 0.12) | (0.13 0.13) |
| 15) 57.290344±0.3222±0.0045 | (0.07 0.07) | (0.07 0.07) | (0.12 0.12) | (0.06 0.06) | (0.12 0.12) | (0.09 0.09) |
| 16) 88.2 | (1.96 0.79) | (1.85 0.74) | (1.92 0.74) | (1.71 0.66) | (2.20 0.88) | (1.83 0.68) |
| 17) 165.5 | (0.19 0.20) | (0.15 0.18) | (1.01 0.29) | (0.20 0.12) | (1.30 0.43) | (0.60 0.13) |
| 18) 183.31±7 | (0.13 0.13) | (0.13 0.13) | (0.07 0.09) | (0.12 0.12) | (0.24 0.07) | (0.15 0.15) |
| 19) 183.31±4.5 | (0.12 0.12) | (0.11 0.13) | (0.10 0.10) | (0.10 0.10) | (0.06 0.07) | (0.12 0.12) |
| 20 ) 183.31±3.0 | (0.11 0.11) | (0.11 0.11) | (0.09 0.09) | (0.09 0.09) | (0.08 0.09) | (0.11 0.11) |
| 21) 183.31±1.8 | (0.09 0.09) | (0.10 0.10) | (0.09 0.09) | (0.08 0.08) | (0.09 0.09) | (0.10 0.10) |
| 22) 183.31±1 | (0.09 0.09) | (0.09 0.09) | (0.09 0.09) | (0.08 0.08) | (0.10 0.10) | (0.09 0.10) |





**Appendix B: Expected differences between PWR19 and PWR22**

Spectroscopic parameters are continuously investigated, updating their values and uncertainty. With respect to PWR19 used here, values for several parameters have been updated in the PWR release of Jan 2022 (PWR22, available at http://cetemps.aquila.infn.it/mwrnet/lblmrt_ns.html). Differences between $T_B$ computed with PWR22 and PWR19 in the 10- 800 GHz range (50 MHz resolution) are reported in Figure B1 for the six typical climatology conditions (tropical, midlatitude summer, midlatitude winter, sub-arctic summer, sub-arctic winter, U.S. standard). The most significant differences with respect to PWR19 are (i) in the 50-70 GHz range and around 118 GHz, due to the update of $O_2$ line-coupling parameters, which now include second order line mixing (Makarov et al. (2020)), (ii) around 183 GHz for the introduction of speed-dependent line shape at this water vapor line (Koshelev et al. (2021)), (iii) above 600 GHz for the inclusion of four water vapor lines (860, 970, 987, 1097 GHz), and (iv) for updating line parameters taken from HITRAN according to the latest release available (HITRAN2020) (e.g. $O_2$ 16O18O isotopologue line at 234 GHz).

Assuming PWR22 as the reference for the most updated spectroscopy, additional uncertainty could be associated to the PWR19 calculations as the typical systematic and random difference with respect to PWR22. These differences have been investigated through a set of diverse atmospheric profiles. The set of 83 atmospheric profiles were selected to represent the diverse range of possible atmospheric conditions (Matricardi (2008)) and it is commonly used to train the regression coefficients in RTTOV (Saunders et al. (2018)). It has also been used extensively in Turner et al. (2022) (e.g., their Appendix A). The spectral difference of PWR22-PWR19 using the diverse profiles, as well as their mean and std, are shown in Figure B2. Note that std difference spectrum stays within the uncertainty calculated for PWR19 (see Fig. 7), and thus it is consistent with that. The only feature for which the std difference is larger than the PWR19 uncertainty is at 234 GHz, related to the $O_2$ 16O18O isotopologue line, for which the strength was lowered by a factor of 4 starting from HITRAN 2016 on. Note that the only channels that are affected are ICI ch. 4 (234 GHz) and MWS ch. 24 (229 GHz), with an impact not larger than $0.13 - 0.26 K$, as can be seen from Table B1, where the convolutions of mean and std difference spectra on instrument channels are reported.





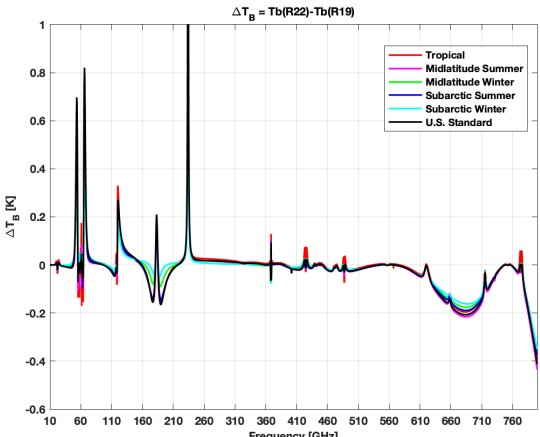

**Figure B1.** Differences between Brightness Temperature computed using PWR22 and PWR19 absorption models (PWR22 minus PWR19). Six typical climatology conditions are considered (tropical, midlatitude summer, midlatitude winter, sub-arctic summer, sub-arctic winter, U.S. standard)

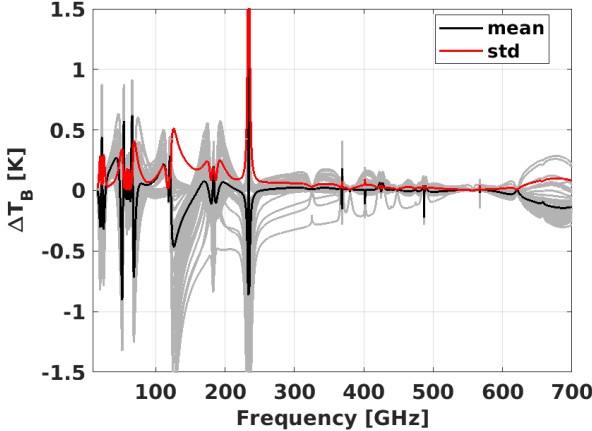

**Figure B2.** Brightness Temperature difference (PWR22 minus PWR19) using 83 diverse profiles (grey lines), and their mean (black) and std (red). The y-axis is limited to $\pm 1.5$K, but the std at 234GHz overreaches 10K.



**Table B1.** Estimated additional systematic and random uncertainty associated with PWR19 calculations, taking latest model release as reference (PWR22).

| Channel (Uncertainty): | MWS $(\delta_{BT}\ \sigma_{BT})$ | ATMS $(\delta_{BT}\ \sigma_{BT})$ | MWI $(\delta_{BT}\ \sigma_{BT})$ | ICI $(\delta_{BT}\ \sigma_{BT})$ | ISMAR $(\delta_{BT}\ \sigma_{BT})$ | MARSS $(\delta_{BT}\ \sigma_{BT})$ |
|---|---|---|---|---|---|---|
| 1 | (-0.23 0.21) | (-0.23 0.21) | (-0.18 0.19) | (-0.01 0.22) | (0.14 0.07) | (0.05 0.07) |
| 2 | (0.14 0.03) | (0.14 0.03) | (-0.22 0.21) | (-0.10 0.09) | (0.16 0.06) | (-0.03 0.18) |
| 3 | (-0.53 0.33) | (-0.53 0.33) | (0.14 0.03) | (-0.06 0.13) | (0.13 0.07) | (0.05 0.17) |
| 4 | (-0.06 0.21) | (-0.87 0.33) | (-0.53 0.33) | (-0.04 0.13) | (0.04 0.14) | (-0.11 0.09) |
| 5 | (0.37 0.17) | (-0.06 0.21) | (-0.06 0.21) | (0.02 0.05) | (-0.10 0.31) | (-0.01 0.22) |
| 6 | (0.50 0.15) | (0.50 0.15) | (0.36 0.17) | (0.01 0.04) | (-0.04 0.13) | |
| 7 | (0.50 0.14) | (0.35 0.12) | (0.50 0.14) | (0.01 0.03) | (0.01 0.03) | |
| 8 | (0.35 0.12) | (0.10 0.05) | (0.05 0.07) | (0.01 0.02) | (0.01 0.04) | |
| 9 | (0.10 0.05) | (-0.04 0.04) | (0.01 0.16) | (0.01 0.01) | (0.02 0.05) | |
| 10 | (-0.04 0.04) | (-0.01 0.04) | (0.13 0.06) | (0.01 0.01) | (0.04 0.04) | |
| 11 | (-0.01 0.04) | (-0.01 0.02) | (0.16 0.07) | (-0.12 0.09) | (0.06 0.04) | |
| 12 | (-0.01 0.02) | (-0.01 0.03) | (0.15 0.07) | | (0.04 0.03) | |
| 13 | (-0.01 0.03) | (0.01 0.03) | (0.05 0.19) | | (0.01 0.01) | |
| 14 | (0.01 0.03) | (0.03 0.03) | (-0.01 0.22) | | (0.01 0.01) | |
| 15 | (0.03 0.03) | (0.03 0.03) | (-0.03 0.19) | | (0.01 0.02) | |
| 16 | (0.03 0.03) | (0.05 0.07) | (-0.07 0.14) | | (-0.12 0.09) | |
| 17 | (0.05 0.07) | (0.05 0.19) | (-0.10 0.10) | | | |
| 18 | (0.05 0.19) | (-0.01 0.22) | (-0.06 0.13) | | | |
| 19 | (-0.01 0.22) | (-0.08 0.12) | | | | |
| 20 | (-0.08 0.12) | (-0.11 0.09) | | | | |
| 21 | (-0.11 0.09) | (-0.05 0.14) | | | | |
| 22 | (-0.05 0.14) | (0.04 0.17) | | | | |
| 23 | (0.04 0.17) | | | | | |
| 24 | (-0.11 0.26) | | | | | |



*Author contributions.* DC, SF, VM, and DG designed the research. DG and DC lead the analysis and data processing, and wrote the original
manuscript. ET, PR, MYT, SL, and FR provided advice and contributed to data analysis. SF and VM procured funding. All the co-authors
helped to revise the manuscript.

*Competing interests.* The authors declare that they have no conflict of interest.

*Acknowledgements.* The authors acknowledge the support of EUMETSAT through the study on "atmospheric absorption models using
ISMAR data" (under contract EUM/CO/20/4600002477/VM). Results are also instrumental for the EUMETSAT VICIRS study (contract
EUM/CO/22/4600002714/FDA).



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
