# Peer review of "Uncertainty of simulated brightness temperature due to sensitivity to atmospheric gas spectroscopic parameters"

_EGUsphere, 2023_

## Author Comment (AC1)

*We thank the reviewer for the positive comments and for recommending publication of our manuscript. Please find below the response to your points.*

**Specific comments**

A) Where the tables 1-3 indicate a range of estimated uncertainty for a parameter, what uncertainty value was used in the sensitivity analysis? (This should be described in the text).

*Indeed, we agree this should be specified. Uncertainty ranges are indicated only when the uncertainty value of the spectroscopic parameter depends on the specific resonant line. We have now added a sentence in Section 2.1 line 163 to clarify this:*

*"It shall be noted that uncertainty ranges are indicated in Tables 1-3 when the uncertainty value of the spectroscopic parameter depends upon the specific resonant line."*

B) Eqn 4: should the limit be as "\nu_0 -> 0" rather than "\nu -> 0"?

*Thank you for spotting this notation inconsistency.*

C) Lines 237-239. This sentence is confusing to me. I read it as saying that most channel passbands are at least 100 MHz from line centres (OK so far), and then it goes on to say that this is true for ICI channels at 325.15+/-1.5 GHz (sidebands 1600 MHz wide) and 664+/-4.2 GHz (sidebands 5000 MHz wide). In these examples, the sideband widths are larger than the offset from the central frequency meaning they overlap around the central frequency, so surely these bands do include the spectral line centre?

*Thank you for this comment. We agree this statement might be misleading, we have now rephrased it as follows.*

*"It shall be noted that all double-sided channels in Table 4, but MWI channel 13, have half-bandwidth smaller than the detuning from the line center, with passbands at least 700 MHz away from any line center, and thus far from the range impacted by the impulse. Similarly, MWI*

*channel 13 is not affected by the impulse because it is located away from any resonant line absorption."*

**Typographical**

D) Line 21: "RTM represents" -> "RTM represent" (consistent use of RTM == radiative transfer models plural).

*We have changed it accordingly.*

E) Line 71: "from 183 GHz and 664 GHz" -> "from 183 GHz to 664 GHz".

*We have changed it accordingly.*

F) Inconsistent use of "vapour" and "vapor" throughout the text.

*We have now opted for "vapour" throughout the paper.*

---

## Author Comment (AC2)

*We thank the reviewer for the positive comments and for recommending publication of our manuscript. Please find below the response to your points.*

A) The title should be expanded to indicate that this study is focused on the microwave to submillimeter. (A trivial change)

*The title has been modified as follows:*
*Uncertainty of simulated brightness temperature due to sensitivity to atmospheric gas spectroscopic parameters from centimeter- to submillimeter-wave*

B) The authors clearly understand the importance of accounting for the covariance in the uncertainties between different parameters. The easiest example to state is that the uncertainties in Cf and Cs (the continuum coefficients) are anti-correlated; this has been well known for years and indeed was discussed. However, this leads to two concerns:

1. While the paper alludes to accounting for these covariances, there is no explicit statement on how this was determined and what those covariances were assumed to be in this analysis. This could be addressed with two additional tables (one for water vapor, one for oxygen) that provide correlation values between parameters (and when connected with table 1 or table 2 could be converted by the reader (like me) into covariances). I understand that there could be (and probably is) some spectral variability between the correlation of any two parameters, but still even having mean values of the correlations would be useful.

*The covariance values are reported in the supplement material: this was mentioned in the "code and data availability" but we missed to mention also within the main text. We have now added an explicit reference to this at the beginning of Sec.4, where we also discuss the method used to determine the covariances:*

*"The full uncertainty covariance matrix Cov(p), as well as the correlation matrix Cor(p), for the set of 135 dominant spectroscopic parameters for water vapor and oxygen absorption is provided in the form of supplement material along with the manuscript."*

*Since these matrices are 135x135, we deem as impractical to report them explicitly within the main text.*

2. One of the most fascinating plots in the Cimini et al. 2018 paper was Figure 9, which showed the covariance in the resulting Tb calculation from the spectroscopic parameter uncertainties. There is no information like this in this current paper, and it is essential before it be accepted for publication.

Agreed. We have now added the requested figure in Appendix A, along with the following statement:

"We also show in Figure A1 a graphical representation of the full covariance matrix of Tb uncertainties for MWI, ICI, MWS and ATMS, relative to horizontal polarisation and US standard climatology (see supplement material for other climatologies and vertical polarisation)."